# Digital finance and performance in innovation: Evidence from universities

**Song Tan[1]◉, Lan Tao[2]◉ \***

**1** School of Economics and Management, Huainan Normal University, Huainan, China, **2** School of Economics and Management, China University of Geosciences, Wuhan, China

◉ These authors contributed equally to this work.
\* lantao@cug.edu.cn

## Abstract

This study evaluated 72 universities' performance innovation during 2011 to 2019 of panel data, using the data envelopment analysis–Malmquist method. The study used benchmark regression to analyse the relationship between digital finance and the universities' innovation performance. The aim was to improve innovation performance and promote national innovation across countries. According to the results of the empirical analysis, digital finance positively affects innovation performance. That finding was confirmed through advanced robustness test evaluation, such as limited information maximum likelihood, two-stage least squares, and interactive fixed effects. Moreover, based on information theory, the digital finance influence mechanism improves credit demand and financial efficiency. Additionally, innovation performance survived spatial overflow effects. Lastly, the paper concludes with some implications for improving digital financial coverage and constructing innovation networks among universities.

## 1. Introduction

Product innovation and the implementation of key processes are becoming vital resources for countries to gain a competitive advantage [1, 2]. Hence, educating future entrepreneurs is seen to promote individual countries' innovation performance [3]. Innovation education in university originated in the United States. After the second world war, the United States entered an age of rapid economic development, prosperity, and highly developed industrial civilization. To meet the demands brought on by post-World War 2 economic development, colleges and universities began to explore entrepreneurship and innovation education, which can help students improve their innovation knowledge and achieve human capital effects in their country [4]. Regarding the effectiveness of innovation education, Anoosheh found that it positively affects students' entrepreneurship goals and enhances social skills [5].

In China, entrepreneurial talent is key to the development of an entrepreneurial economy. Hence, achieving internal development and cultivating innovative and entrepreneurial talent to meet current needs have become key to the reform and development of higher education [6]. However, innovation in colleges and universities is a high-risk activity that is associated with high costs and uncertain output, especially in an economic downturn. The financial

**Data Availability Statement:** The authors declare that data come:http://www.stats.gov.cn/zs/tjwh/tjzl/202302/t20230220_1913734.html.

**Funding:** This study is supported by Anhui Philosophy and Social Science Planning Project (Project approval No: AHSKQ2020D75).

**Competing interests:** The authors have declared that no competing interests exist.

constraints universities face is a key factor restricting the improvement of innovation technology and patents. Strategies to improve universities' performance in innovation constitute a main research problem related to achieving internal development.

Stable and sufficient financial resources are important for the sustainable promotion of innovation activities in colleges and universities. Furthermore, efficient, and low-cost financial services are key elements for high-quality output in innovation and entrepreneurship. Traditional finance faces the problems of attribute, stage, and field mismatch in the process of serving enterprises' production, and financing difficulty and expensive financing have always been problems restricting innovation in universities [7].

In recent years, along with the rapid evolution of artificial intelligence, blockchain, cloud computing, big data, and other emerging technologies, digital technology is continuously being integrated into the financial industry. Furthermore, digital finance, as an emerging financial model, provides great development space to effectively reduce financial transaction costs, broaden the scope of financial services, and improve financial access; it also provides important support for innovation education in universities. In this context, how to use digital finance to improve universities' performance in innovation is not only a major application problem that has been pushed to the forefront of the reform but also a theoretical problem that needs to be studied.

This study examined the impact of digital finance on universities' performance in innovation to draw useful insights and references to strengthen the theoretical research on and the practical application of innovation in universities through empirical methods. Compared with the existing literature, this paper's contributions are as follows. We are the first to focus on the digital finance perspective. Specifically, we pioneered exploration of the relationship between digital finance and universities' performance in innovation and entrepreneurship. Another contribution lies in the fact that this study empirically investigated the digital finance mechanism's relationship with university-level performance in innovation and entrepreneurship. In this study, we reconstructed the university-level innovation performance index using the data envelopment analysis–Malmquist method. Finally, this paper provides policy recommendations to better assess the level of development of innovation in universities and improve national levels of innovation and technology.

This paper is organized as follows. Section 2 describes related literature on digital finance and innovation performance. Section 3 describes the data source and model, with four subsections presenting the empirical results, evaluation of innovation performance, and the relationship between digital finance and innovation performance. Section 4 describes the robustness test including an endogeneity test and a test in which the variables were changed. Section 5 discusses the influence mechanism of digital finance on universities' innovation performance. Section 6 concludes and provides policy implications and future research directions.

## 2. Literature review and hypothesis

### 2.1 Digital finance and innovation performance

Innovation is one of the driving factors of economic and social development, and the Austrian economist Schumpeter's "innovation theory" defines innovation as a new combination of production factors. It is an effective way to cultivate innovative talents [8]. The training of innovative talents is inseparable from financial support, and there is an important economic relationship between digital finance and the performance of innovation in colleges and universities.

In view of the uneven and insufficient development of traditional financial system, the long fund flow cycle of universities and the insufficient supply of funds to universities in less

developed areas restrict the improvement of innovation performance [9]. The vigorous development of information technology has given birth to digital finance as an emerging financial service model. Digital inclusive finance has a positive impact on financial inclusion and stability, provides a basis for innovation, and can effectively increase the investment in innovation in universities. Combing the existing studies [10, 11].

The impact of digital inclusive finance on the performance of university innovation can be summarized as follows: digital finance can accurately assess the default risk of customers through big data modeling, and carry out real-time and dynamic supervision of credit customers, effectively control and identify risks, avoid adverse selection and moral risks in the financial market, improve risk control ability, explore more business space, and bring more new opportunities for university innovation. It brings more new opportunities for university innovation [12]. At the same time, digital finance also effectively reduces the cost of information search and matching between borrowers and lenders. Goldfarb & Tucker pointed out that big data can effectively reduce economic costs and help improve the performance of university innovation and entrepreneurship [13]. Finally, the development of digital finance provides a good financial environment for university innovation and entrepreneurship. Trinugroho suggest that Digital inclusive finance has bred a distributed business pattern with strong inclusiveness and permeability, which is conducive to the communication and cooperation of innovation subjects in the distributed innovation network, constructing a collaborative innovation mechanism and helping to improve the quality of university innovation and entrepreneurship [14].

In addition, the combination of traditional finance such as payment, investment and credit with digital technology effectively improves the convenience of customers using financial services and provides a good external environment for financial services to improve the performance of university innovation and entrepreneurship. In summary, this study proposed the following research hypothesis 1.

*H1*: *Digital finance can effectively promote the improvement of innovation and entrepreneurship.*

## 2.2 Mechanism of innovation and entrepreneurial performance

Based on the existing literature and drawing on signalling theory, this study hypothesized that digital finance influences university-level innovation performance through the following two channel mechanisms.

First, the development of digital finance can improve the quality of university-level innovation by expanding credit supply and relieving the financing constraints universities face [15]. Entrepreneurial returns are uncertain, and information asymmetry tends to favour capital owners. Potential entrepreneurs are bound to face financing constraint dilemmas, and university innovation activities are also affected by uncertain financing constraints [16]. Relying mainly on borrowers' collateral materials and financial information to decide whether to lend renders the decision-making basis too narrow, which contributes to the financing difficulty dilemma. The development of digital inclusive finance can facilitate the use of big data, machine learning, and other methods to collect and analyse borrowing individuals' bills, shopping records, and other information to expand the credit supply, relieve financing difficulties and expensive financing for universities [17], and help improve the quality of innovation and entrepreneurship.

Second, the development of digital finance can also effectively improve financial efficiency and inject new momentum towards improving the quality of innovation [18]. Financial

efficiency is an important dimension to measure financial institutions' effectiveness at promoting the allocation of funds in economic activities. Digital finance's greatest functions are to overcome temporal and spatial limitations and establish a direct point-to-point contact between financial product providers and demanders, thus accelerating the speed of capital circulation across time and space, which will, in turn, enhance financial efficiency and reduce financial enterprises' operating costs [19]. Tang suggested that digital finance can also, to a certain extent, drive the reshaping of the traditional financial system, force the traditional financial sector to transform and upgrade, and improve the allocation efficiency and risk control ability of financial resources [20]. Moreover, the development of digital finance lowers the financial services threshold, improves financial efficiency, and contributes significantly to regional total factor productivity. In summary, this study proposed the following research hypothesis 2.

*H2*: *Digital finance can, by expanding the credit supply and improving financial efficiency, promote universities' innovation performance.*

According to the abovementioned literature, signalling theory fits well with the current study's objective regarding university-level digital finance and innovation performance. It is thought that digital finance can, by decreasing market moral hazards and enhancing the efficiency of financial services, improve universities' innovation performance [17]. Furthermore, digital finance lowers the financial services threshold, improves financial efficiency, and contributes significantly to regional total factor productivity [19]. Based on the above discussion, the present study's conceptual model is visualized in Fig 1.

The above digital finance and innovation performance model encompasses digital finance, credit supply, and innovation behaviours in higher education. The figure shows two mechanisms. First, the development of digital inclusive finance can facilitate the use of big data and machine learning to collect and analyse a large amount of organizational borrowing information to expand the credit supply, relieve the financing difficulties and expensive financing universities face [17], and help improve the quality of innovation and entrepreneurship. Secondly, the development of digital finance will lower the financial services threshold, improve financial efficiency, and significantly contribute to higher education innovation and entrepreneurship.

Although the present study has contributed to the accumulation of innovation education research, it has several limitations. First, the research lacks theoretical depth; specifically, it

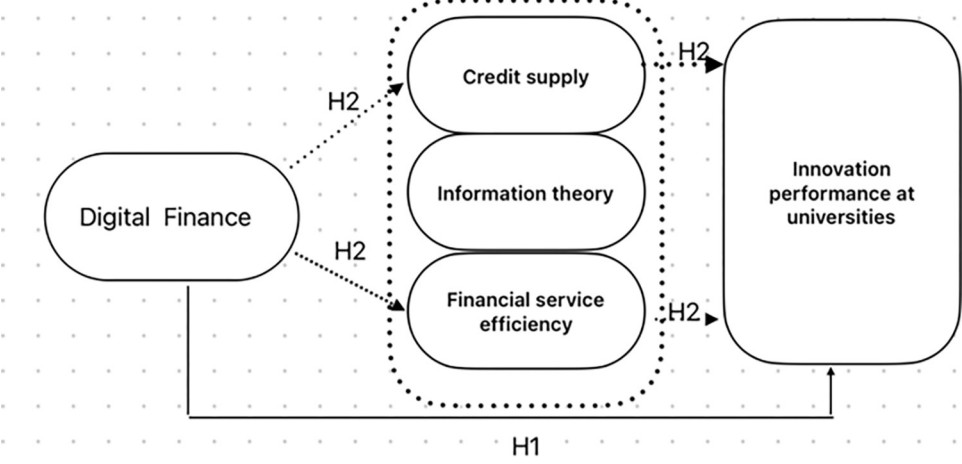

**Fig 1. Digital finance conceptual model.**

lacks scientific performance evaluation theoretical models, a reasonable index design, and comprehensive evaluation content. Second, the evaluation subjects were relatively homogenous as the study mainly focused on teachers and students. Third, regarding the research methods, the study was mainly qualitative, and the quantitative research lacks large data samples; consequently, the research conclusions are neither convincing nor universally applicable. Nevertheless, compared with the extant literature, we drew upon information theory and the two perspectives of credit supply and financial service efficiency to discuss the relationship of digital finance and university-level innovation performance with the aim of improving universities' innovation development and addressing the problem of insufficient investment in university-level innovation and entrepreneurship.

## 3. Data source and model setting

To empirically test the effect of digital finance on universities' innovation performance, this study constructed an econometric model of the influence of digital finance on the quality of innovation in universities. This section will explain the reason that innovation performance input and output data sources, measuring of innovation performance and baseline model.

### 3.1 Data sources

The index data related to colleges and universities used in this study were mainly obtained from the 2010–2019 Compilation of Basic Statistics on Colleges and Universities of the Ministry of Education, and the city-level data were obtained from the China City Statistical Yearbook for the year's corresponding to the period of interest. The final sample comprised 72 colleges and universities' 10-year panel data.

The independent variable was innovation performance. The extant literature suggests the convenience of using human capital, financial investment, and education Material indicators to closely examine higher education innovation capabilities [21] through DEA analysis. Hence, in this study, based on the existing literature [22], we divide the university innovation inputs into three major inputs of Innovation Human Capital, Financial investment of Innovation Education, and Innovation Education Material in Table 1.

In terms of outputs indicators for innovation performance, this paper based on Xiong [23], to measure university innovation performance outputs into three major outputs of innovation talent, innovation scientific research and innovation education effects. Considering that colleges and universities are the main places for innovation knowledge production and innovation talent cultivation, the main work of their workers can be divided into number of innovation activities administrative staff and innovation teaching and support staff two items, we must keep in mind that the capacity a university must innovate depends on talent assets and knowledge it possesses [24],which means this perspective of innovation performance depends on innovation human capital, such as innovation activities administrative staff and innovation teaching and support staff [25].

In terms of financial investment of innovation education. This paper refers to the general division method of income sources of colleges and universities and divides the financial input of colleges and universities into three items: state innovation education financial allocation, innovation education business income and other innovation activities income. Among them, the state innovation education financial allocation includes education scientific innovation research fund appropriation; the innovation education business income of colleges and universities mainly refers to the tuition fee income of innovation colleges; other innovation activities income refers to other non-financial and non-education business innovation income of colleges and universities, such as innovation business income, contribution from innovation

**Table 1. Construction of innovation performance index system in universities.**

| Input Indicators | | | | General (8 items) |
|---|---|---|---|---|
| Innovation Human Capital | 1 | | Number of innovation activities administrative staff | Y |
| | 2 | | Number of innovation teaching and support staff | Y |
| Financial investment of Innovation Education | 3 | | State innovation education financial allocation | Y |
| | 4 | | Innovation Education Business Income | Y |
| | 5 | | Other innovation activities income | Y |
| Innovation Education Material | 6 | | The value Of Innovation research instruments and Equipment | Y |
| | 7 | | Number of books about innovation research | Y |
| | 8 | | Fixed asset of innovation research | Y |
| Output Indicators | | | | General (11 items) |
| Innovation Talent | 1 | | Innovative entrepreneurial Projects | Y |
| | 2 | | Changjiang Scholar Award Program | Y |
| | 3 | | Number of International Students | Y |
| Innovation Scientific Research | 4 | | Number of Patent conversions | Y |
| | 5 | | Number of published papers (web of science) | Y |
| | 6 | | Number of Patents | Y |
| | 7 | | Number of innovating awards | Y |
| Innovation Education Effects | 8 | | Number of consultancy studies adopted | Y |
| | 9 | | Technology transfer contract amount | Y |
| | 10 | | Total amount of patent sales | Y |
| | 11 | | Number of further education and training | Y |

Note: Y represents yes, those indicators both put into the model.

affiliated units [26]. Considering that financial investment of innovation education play a critical role in innovation development, especially for the development of innovation research and patent promotion in universities [27].

Innovation Education Material. Innovation education effects is an important indicator of innovation performance [28]. In his work propose a model of innovation performance where the probability of being innovative performance depends on the innovation education effects. They examine whether the technology transfer, patent and consultancy adopted can explain different types of innovation performance at universities. Dutta analyze the relationship between innovation material input and innovation education effects [29].

Therefore, the innovation material input of colleges and universities mainly includes three aspects: The value of innovation research instruments and equipment, number of books about innovation research, and fixed asset of innovation research. Based on numerous literatures [30, 31], this paper also choose three output indicators include innovation talent, innovation scientific research, and innovation education effects. According to Marta [31], we measure innovation performance from universities through DEA-Malmquist methods. Research has addressed the application of the DEA-Malmquist in innovation performance at universities, considering non-profit organizations focused on innovation production rather than profitability [32]. These measures respond better to universities innovation performance, and the system of innovation performance indicators of colleges and universities is shown in Table 1.

Dependent variable: digital finance. This paper follows the existing studies and adopts the Digital Finance Index released by the Digital Finance Research Center of Peking University to measure the development of digital finance in China. The Digital Inclusive Finance Index contains three first-level dimensions: breadth of coverage, depth of use and digitization, and the index provides data material for relevant empirical studies in the field of digital finance.

Other control variables: With reference to the existing literature [33], the following four factors, which are closely related to innovation performance, are used as control variables in this paper. The level of fixed asset investment (INV), which is measured by the ratio of total social fixed asset investment to regional GDP; labor input (LABOR), which is expressed by the number of urban employees; industrial structure (IS), which is expressed by the proportion of added value of tertiary industry to regional GDP; and economic development level, which is expressed by per capita GDP.

## 3.2 Model setting

Based on Kubak [34], considering universities is non-profit organizations, innovation performance has to reflect the dynamic innovation change status of relative innovation efficiency, therefore we adopt Malmquist index to better analysis innovation performance chance from panel data.

Malmquist Index Model of Innovation Performance in Higher Education: the concept of Malmquist index was proposed by Malmquist in 1953's, after calculating the index using the DEA method, it was decomposed and quantified into two aspects [35]: first, the technical efficiency change of the e valuate DMU in two periods, and second, the change in production technology (TC), i.e., the change in the production frontier in the DEA analysis. Technological Change (TC). The change in the production frontier in the DEA analysis, calculated by the following equation:

$$M^t = \frac{D_C^t(x^{t+1}, y^{t+1})}{D_C^t(x^t, y^t)} \tag{1}$$

The equal (1) defined as first period technology change, $D_C^t(x^{t+1}, y^{t+1})$ represents the $(x^t, y^t)$ period of $t$ distance function; and $D_C^t(x^t, y^t)$ represent the $(x^{t+1}, y^{t+1})$ period of $t$ distance function.

$$M^{t+1} = \frac{D_C^{t+1}(x^{t+1}, y^{t+1})}{D_C^{t+1}(x^t, y^t)} \tag{2}$$

The equal (2) represent the second period technology change, $D_C^{t+1}(x^{t+1}, y^{t+1})$ represents the $(x^{t+1}, y^{t+1})$ period of $t+1$ distance function; and $D_C^{t+1}(x^t, y^t)$ show the $(x^t, y^t)$ distance function on the $t+1$ periods.

$$M(x^t, y^t, x^{t+1}, y^{t+1}) = (M^t \times M^{i+1})^{\frac{1}{2}} = \left[\frac{D_c^t(x^{t+1}, y^{t+1})}{D_c^t(x^t, y^t)} \times \frac{D_c^{t+1}(x^{t+1}, y^{t+1})}{D_c^{t+1}(x^t, y^t)}\right]^{\frac{1}{2}} \tag{3}$$

The equal (3) defines the two steps Malmquist index, according to the equal (1) and equal (2), which to be decomposed into a composite technical efficiency change index and a technical progress index.

We evaluate performance innovation based on the equal (3). Then to analysis the relationship between digital finance and performance of innovation through baseline model.

Based on the previous theoretical analysis and hypotheses, this paper constructs the following benchmark model to test the impact of digital finance on the performance of innovation in

universities, the model as following:

$$\ln innov_{it} = \alpha_0 + \alpha_1 \ln df_{it} + \gamma \, control_{it} + \mu_i + v_t + \varepsilon_{it} \tag{4}$$

The equal (4), $i$ and $t$ denote city and year, respectively; innov denotes the independent variable (performance of innovation in universities); and the $df$ defines the dependent variable (digital financial index); the control denotes a set of relevant control variables, α0, α1 and γ denote the coefficients represented α0, α1 and γ; and $\mu_i$ is used to control for university city fixed effects and $v_t$ is used to control for time fixed effects; $\varepsilon_{it}$ denotes the random error term.

## 4. Empirical results

Based on the econometric model constructed in the previous section, this section first conducts a benchmark regression analysis to test the impact of digital inclusive finance on the quality of innovation and entrepreneurship, and to effectively deal with possible endogeneity issues. A series of robustness tests are also conducted to ensure that the results are robust and reliable.

### 4.1 Innovation performance in higher education

The innovation performance from 2010 to 2019 is calculated based on equal (3), and the results are shown in Table 2. The Table 2 reports that the mean value of Malmquist index of innovation performance of universities is 1.038, which is greater than 1. The results imply that the input and output performance of universities in general shows an upward trend during the 10 years, and the resource allocation shows an optimized state.

The mean value of comprehensive efficiency is 0.997, it lower than 1, which shows that universities have a decreasing trend in the comprehensive efficiency of resource allocation during the 10 years. Specifically, the mean value of pure technical efficiency is 0.9, which is less than 1, and the scale efficiency is 0.9, which is also less than 1. This indicates that the decrease of management technical efficiency and the deterioration of scale efficiency of resource allocation both bring negative effects on the efficiency of universities' innovation performance. Finally, the mean value of management innovation is 1.068, which is greater than 1, indicating that the innovation in the management system of universities conducive to stimulating the innovation input-output performance and achieving the optimal improvement of resource allocation.

In additionally, management innovation has become the main driving force for the improvement of input-output performance of universities from 2010 to 2019.

**Table 2. Malmquist index results of innovation in universities over 2010 to 2019.**

| Year | Pure technical efficiency | Scale efficiency | Comprehensive efficiency | Management Innovation | Malmquist Index |
|---|---|---|---|---|---|
| year | PTE | SE | EC | TC | MI |
| 2010–2011 | 1.017 | 1.003 | 1.015 | 1.014 | 1.022 |
| 2011–2012 | 0.972 | 1.018 | 0.999 | 1.022 | 1.029 |
| 2012–2013 | 1.042 | 1.007 | 1.050 | 0.971 | 1.020 |
| 2013–2014 | 0.989 | 1.013 | 1.011 | 1.050 | 1.062 |
| 2014–2015 | 0.912 | 0.942 | 0.868 | 1.207 | 1.048 |
| 2015–2016 | 1.041 | 0.917 | 0.964 | 1.252 | 1.015 |
| 2016–2017 | 0.877 | 0.787 | 0.855 | 1.374 | 1.144 |
| 2017–2018 | 0.972 | 1.047 | 1.115 | 0.902 | 0.994 |
| 2018–2019 | 1.019 | 1.035 | 1.096 | 0.826 | 1.004 |
| Average value | 0.982 | 0.975 | 0.997 | 1.068 | 1.038 |

**Table 3. Baseline regression results of the impact of digital finance on the performance of innovation.**

| Variables | Model 1 | Model 2 | Model 3 | Model 4 | Model 5 | Model 6 |
|---|---|---|---|---|---|---|
| lndfi | 0.2125*** (0.1016) | 0.2177*** (0.1134) | 0.2891*** (0.1043) | 0.2921*** (0.1050) | 0.3012*** (0.1061) | 0.3071*** (0.1055) |
| invest | | | 0.1372* (0.1131) | 0.1334* (0.1123) | 0.1297* (0.1144) | 0.1127 (0.1246) |
| lnlabor | | | | 0.0843* (0.0489) | 0.09* (0.0493) | 0.0816* (0.0493) |
| IS | | | | | 0.0010 (0.0015) | 0.0013 (0.0015) |
| lnpgdp | | | | | | 0.1241* (0.0690) |
| _cons | 2.0199*** (0.5728) | 6.44*** (1.0741) | 6.10*** (1.0968) | 5.417*** (1.2219) | 5.4572*** (1.2230) | 3.4527** (1.4944) |
| Observations | 720 | 720 | 720 | 720 | 720 | 720 |
| Time Effect | Y | Y | Y | Y | Y | Y |
| Individual effects | Y | Y | Y | Y | Y | Y |
| R-squared | 0.8679 | 0.8729 | 0.8731 | 0.8772 | 0.8789 | 0.8728 |

Note

*, **, *** denote significant at the 10%, 5%, 1% significance levels, respectively; robust standard errors of clustering at the university level are in parentheses; same below.

## 4.2 Digital finance and the performance of innovation

To evaluate the relationship between digital finance and university innovation performance, we based on the equal (4) do the benchmark regression, the regression results are reported in Table 3.

The estimation results of models 1–6 in Table 3, presents that the coefficients and significance of the core explanatory variables and control variables do not change significantly, which indicates that the estimation results have strong robustness. And the coefficients of the core explanatory variables in the estimation results of each model, we can see that the coefficients of digital finance are significantly positive at the 1% significance level, which indicates that digital finance helps to improve the quality of innovation; at the same time, this also initially verifies the validity of hypothesis H1.

The estimation results of control variables show that labor input and urban economic development status can significantly contribute to the performance of university innovation and entrepreneurship. Labor is the most important input factor for innovation in universities, and the success of entrepreneurship depends largely on the availability of suitable labor. The better the economic development, the more complete the factor market and the better the market mechanism, which can provide advanced and comprehensive factor supply and sound market mechanism for innovation activities. Although the coefficient of fixed asset investment is positive, it fails to pass the significance test. The possible reason is that fixed asset investment is the most rapid and direct way to stimulate the local economy, and under the pressure of GDP competition, local governments are more inclined to invest in infrastructure construction. The coefficient of industrial structure variable fails to pass the significance test, which may be due to the slow development of high-end service industry in the current tertiary industry structure of China, which does not support the innovation of universities enough and fails to effectively improve the innovation performance of universities.

# 5. Endogenous and robust test

## 5.1 Endogenous

In section 4 we empirically testing the impact of digital finance on the innovation performance of universities, both individual and time effects are controlled for, which can effectively solve part of the endogeneity problem caused by omitted variables. To mitigate the endogeneity

**Table 4. Estimation results of instrumental variables.**

| Variables | LIML | 2SLS |
|:---:|:---:|:---:|
| | **Model 1** | **Model 2** |
| *lndfi* | 0.037*** (0.2321) | 0.062** (0.2553) |
| *invest* | 0.0810 (0.1210) | 0.0828 (0.1216) |
| *lnlabor* | 0.0838* (0.0484) | 0.0838* (0.0486) |
| *IS* | 0.0004 (0.0013) | 0.0004 (0.0013) |
| *lnpgdp* | 0.0627 (0.0682) | 0.0623 (0.0686) |
| *_cons* | 3.7580** (1.4208) | |
| Observations | 720 | 720 |
| Time Effect | Fixed | Fixed |
| Individual effects | Fixed | Fixed |
| Kleibergen-Paap rk LM (p-value) | 25.12 (0.0000) | 25.12 (0.0000) |
| Cragg-Donald Wald F | 127.150 | 117.350 |
| R-squared | 0.471 | 0.132 |

Note: Kleibergen-Paap rk LM original hypothesis is that instrumental variables are under identified; Cragg-Donald Wald F original hypothesis is that endogenous variables are weakly identified; Stock-Yogo test for LIML and 2SLS The 10% thresholds were 19.36 and 16.38, respectively.

problem more effectively, this paper then adopts the instrumental variables approach to deal with the endogeneity problem. Referring to the existing literature, this paper adopts Internet penetration as an instrumental variable for digital finance. Internet penetration can effectively reflect the infrastructure development of digital finance development; there is no direct effect of Internet penetration on innovation when controlling for other relevant variables. Therefore, Internet penetration satisfies the requirements of relevance and exogeneity of instrumental variables. In addition, the limited information maximum likelihood (LIML) method is less sensitive to instrumental variables compared to the two-stage least squares (2SLS) method, and the LIML estimates are less affected even in the presence of weak instrumental variables. In making the choice of the instrumental variable estimation method, LIML is used to estimate the instrumental variables in this paper to effectively overcome the possible potential weak instrumental variable problem. To facilitate comparison and ensure the robustness of the instrumental variable estimation results, the estimation results of 2SLS are also presented in this paper. The specific results are reported in Table 4.

Observing the estimated results of instrumental variables reported in Model1 and Model2, there is still a significant effect of digital inclusion on the quality of innovation after mitigating the possible endogeneity problems. The above findings again validate that hypothesis H1 is valid.

## 5.2 Robustness test

To ensure the robustness and reliability of the above research results, this paper further tests the robustness of the benchmark regression in the research methodology, constructs a panel factor error structure model to re-test the impact of digital finance on the performance of university innovation and entrepreneurship. Model 1 reports the interactive fixed effect (IFE) method proposed by Bai for estimation [36], the estimation results are reported in Table 5. In addition, in order to eliminate the effect of extreme values, all continuous variables are subjected to a 0.5% two-sided truncation, and the impact of digital finance on the performance of innovation in universities is tested again, and the estimation results are reported in Model 2.

**Table 5. Robustness test estimation results.**

| Variables | IFE | Truncation processing |
|---|---|---|
| | Model 1 | Model 2 |
| *lndfi* | 0.131*** (0.0286) | 0.1133* (0.1260) |
| *invest* | 0.0356** (0.0223) | 0.0195 (0.0621) |
| *lnlabor* | 0.0474*** (0.0177) | 0.0978** (0.0490) |
| *is* | 0.0003 (0.000) | 0.0219 (0.0016) |
| *lnpgdp* | 0.0408* (0.0279) | 0.0922 (0.0088) |
| *_cons* | 3.3923*** (0.7208) | 3.2114** (1.5132) |
| Observations | 720 | 720 |
| Time Effect | Fixed | Fixed |
| Individual effects | Fixed | Fixed |
| R-squared | 0.452 | 0.617 |

Note

*, **, *** denote significant at the 10%, 5%, 1% significance levels, respectively; robust standard errors of clustering at the university level are in parentheses; same below.

The estimation results are generally consistent with the findings of the previous study. This again verifies that hypothesis H1 is valid.

## 6. Mechanism discussion

### 6.1 The mechanism of credit supply and financial efficiency

From the financial economic theoretical analysis, digital finance may affect the quality of innovation through both expanding credit supply and improving financial efficiency. In this paper, we intend to identify and test the above possible transmission pathways by constructing a mediating effect test model. For credit supply, we refer to Bhunia [37] uses the loan balance per capita, and for financial efficiency. we refer to Huang [38] approach ich uses the loan/deposit (feffi) of financial institutions. Given that this paper uses panel data, the following mediating effects test model is constructed for analysis:

$$lninnov_{it} = \delta_0 + \delta_1 lndf_{it} + \gamma control_{it} + \mu_i + v_t + \varepsilon_{it}$$
$$M_{it} = \beta_0 + \beta_1 lndfi_{it} + \gamma' control_{it} + \mu_i + v_t + \varepsilon_{it} \qquad (5)$$
$$lninnov_{it} = \phi_0 + \phi_1 lndfi_{it} + \phi_2 M_{it} + \gamma'' control_{it} + \mu_i + v_t + \varepsilon_{it}$$

The equal (5), where $M_{it}$ is the possible mediating variable. According to the principle of mediation effect test, if $\delta_1$, $\beta_1$, and $\phi_2$ pass the significance test and the absolute value of the coefficient of $\phi_1$ becomes smaller relative to δ1 or the significance level decreases, it indicates the existence of a mediating effect. $\delta_1$ indicates the total effect size, $\beta_1$ indicates the effect of digital financial inclusion on credit supply and financial efficiency; $\phi_2$ indicates the mediating effect size. Table 6 reports the results of intermediation effect estimation. As shown in Table 6, Model 3 analysis the mechanism of whether credit supply is a transmission pathway for digital finance to affect the performance of innovation in university. The mediation estimation results indicate that credit supply is an effective mediating variable for digital inclusive finance to affect the performance of innovation in university. Digital finance improves the performance of innovation by expanding credit supply and reducing financing constraints. In summary, the mediating effect test indicates that the development of digital finance will contribute to the improvement of innovation quality through expanding credit supply, alleviating financing

**Table 6. Mechanism discussion of credit supply and financial efficiency.**

| Variables | Eq (2) | Eq (3) | Eq (4) | Eq (2) | Eq (3) | Eq (4) |
|---|---|---|---|---|---|---|
| | lninnov | lncredit | lninnov | lninnov | lnfeffi | lninnov |
| | Model 1 | Model 2 | Model 3 | Model 4 | Model 5 | Model 6 |
| lndfi | 0.2170*** (0.1055) | 0.2770*** (0.0619) | 0.2170** (0.011) | 0.2870*** (0.025) | 0.1743** (0.0425) | 0.2651** (0.091) |
| lncredit | | | 0.123** (0.0400) | | | |
| lnfeffi | | | | | | 0.1141** (0.0241) |
| Control variables | Control | Control | Control | Control | Control | Control |
| Observations | 720 | 720 | 720 | 720 | 720 | 720 |
| Time Effect | Fixed | Fixed | Fixed | Fixed | Fixed | Fixed |
| Individual effects | Fixed | Fixed | Fixed | Fixed | Fixed | Fixed |
| R-squared | 0.4271 | 0.3652 | 0.4726 | 0.6723 | 0.545 | 0.4225 |

Note

*, **, *** denote significant at the 10%, 5%, 1% significance levels, respectively; robust standard errors of clustering at the university level are in parentheses; same below.

constraints, and enhancing financial efficiency have positive impacts on the performance of innovation in university. In the process of rapid development of digital finance in China, expanding credit supply and improving financial efficiency are important factors in promoting the performance of innovation in universities. The above research results indicate that hypothesis H2 is valid.

## 6.2 The mechanism of spatial overflow

In the context of talent innovation, all universities are actively deploying to promote innovation education programs, hoping to improve the employment rate of schools, enhance the leading role of science and technology, and take the lead in the new round of competition, which to some extent intensifies the competition for innovation performance among universities. This competition may give rise to the spillover effect of mutual comparison, that is, the improvement of innovation quality in nearby universities will stimulate neighboring regions to make more efforts to promote the development of innovation education to achieve better results in the improvement of innovation performance. In view of this, this paper tries to adopt the classical spatial econometric models—spatial lag model (SLM), spatial error model (SEM) and generalized spatial autoregressive model (SAC)—to test the impact of digital finance on innovation performance again. The spatial lag model is set as follows:

$$\ln \text{innov}_{it} = \rho \sum_{j=1} W_{ij} \text{lninnov}_{jt} + \alpha_1 \ln dfi_{it} + \gamma\, \text{control}_{it} + \mu_i + v_t + \varepsilon_{it} \tag{6}$$

The Eq (5), $\rho$ denotes the spatial lag term coefficient; control variables is consistent with the basic model; $w$ represents the spatial weight matrix, and the spatial distance weight matrix is used here to construct the measurement model, and the matrix elements are set as follows:

$$w_{ij} = \begin{cases} 1/d_{ij}^2 & i \neq j \\ 0 & i = j \end{cases} \tag{7}$$

The spatial error model (SEM) is set as follows:

$$\text{lninnov}_{it} = \alpha_1 \ln df_{it} + \gamma\, \text{control}_{it} + \mu_i + v_t + \varepsilon_{it}$$
$$\varepsilon_{it} = \lambda \sum_{j=1} W_{ij} \varepsilon_{jt} + \mu_{it} \tag{8}$$

**Table 7. Spatial overflow analysis.**

| Variable | SLM | SEM | SAC |
|---|---|---|---|
| | **Model 1** | **Model 2** | **Model 3** |
| *lndfi* | 0.322*** (0.0810) | 0.313*** (0.0822) | 0.296*** (0.0759) |
| *invest* | 0.1125*** (0.0631) | 0.1071** (0.0642) | 0.1091** (0.0585) |
| *lnlabor* | 0.0766*** (0.0376) | 0.0711** (0.0376) | 0.0811 ** (0.0268) |
| *IS* | 0.0011 (0.0014) | 0.0011 (0.0014) | 0.0014 (0.0013) |
| *lnpgdp* | 0.114*** (0.0631) | 0.1234** (0.0640) | 0.1127* (0.0468) |
| $\rho$ | 0.0899* (0.0473) | | 0.2761*** (0.1033) |
| $\lambda$ | | 0.0821* (0.0480) | -0.0227*** (0.1227) |
| *Time fix* | Y | Y | Y |
| *Individual* | Y | Y | Y |
| N | 720 | 720 | 720 |
| R square | 0.372 | 0.411 | 0.392 |

Note

*, **, *** denote significant at the 10%, 5%, 1% significance levels, respectively; robust standard errors of clustering at the university level are in parentheses; same below.

In Eq (7), $\lambda$ defined as spatial autocorrelation, considering the autocorrelation between performance of innovation and entreprenerd in universities, The generalized spatial autoregressive model (SAC), which considers both spatial lag autocorrelation and spatial error autocorrelation, has stronger explanatory power, and the model is set as follows:

$$\text{lninnov}_{it} = \rho\sum\nolimits_{j=1} W_{ij}\text{lninnov int}_{jt} + \alpha_1\ln dfi_{it} + \gamma\text{control}_{it} + \mu_i + v_t + \varepsilon_{it}$$

$$\varepsilon_{it} = \lambda\sum\nolimits_{j=1} W_{ij}\varepsilon_{jt} + \mu_{it} \qquad (9)$$

Table 7 report the estimation results of SLM, SEM and SAC in model 1 to 3, respectively. In Models 1–3, the coefficients of digital significantly positive at the 1% level of significance, which indicates that digital inclusion can still significantly contribute to the performance improvement of innovation under the condition that spatial factors are considered. Observing the estimated values of the spatial lag term reported in Models 1 and 3, they are significantly positive at least at the 10% significance level, which indicate that the improvement of innovation performance in universities has a demonstration effect and will lead to the improvement of innovation performance in neighbor universities. Under the dual pressure of economic development and political promotion, the competitive interaction between neighboring universities will make local governments imitate the innovation policies, and the improvement of the performance of innovation in this universities will also cause neighboring university to pay attention to the performance of innovation and promote the improvement of the performance of innovation in neighboring universities, the performance of innovation has a positive spatial spillover effect. This is undoubtedly beneficial to the promotion and development of innovation and entrepreneur education in China. By strengthening the demonstration effect of inter-university innovation and entrepreneurship, it will steadily improve the performance of innovation in China, enhance the endogenous power of economic development, and boost the high-quality development of education.

## 7 Conclusion and implications

In recent years, digital finance has been flourishing in the world, and the performance of innovation with university is an important driver of high-quality economic development.

We conduct an in-depth theoretical discussion and empirical test on the impact of the development of digital finance on the performance of innovation in university. Combining China's university, the performance of innovation index digital finance index and prefecture-level city statistical yearbook data. We through the bench regression model to empirically analysis the relationship between digital finance and performance of innovation and entrepreneurship. The result find that digital finance can significantly promote the performance of innovation during the 2011 to 2019, and this study conclusion still holds after the endogeneity problem treatment and a series of robustness tests. Second, the mechanism analysis results show that digital finance can influence the performance of university innovation by expanding credit supply and improving financial efficiency. Third, the spatial econometric model study shows that digital finance can still significantly contribute to the improvement of university innovation quality under the consideration of spatial factors, and there is a significant demonstration effect on the improvement of innovation performance.

The research of this paper has the following implications: first, continue to vigorously promote the development of digital finance, enhance the technical level of digital finance, and improve the digital financial service system. The development of digital finance can help improve the performance of innovation in China, and the development of digital finance should be vigorously promoted. By improving the construction of digital infrastructure, encouraging the innovation of diversified financial service business models, expanding the coverage of digital finance with the help of big data, cloud computing and artificial intelligence, and enhancing the depth of digital inclusive finance usage, the overall development of digital finance should be promoted. In addition, considering the current reality that China's digital financial technology is still low, we should increase the investment in innovation of digital financial technology and encourage the innovative application of digital financial technology, to effectively play its role in improving the performance of innovation and entrepreneurship. Secondly, the financial structure and institutional arrangements should be adjusted scientifically and reasonably according to the characteristics of university, and the policies should be tailored to university of different scales. For university, they should further optimize the allocation of financial resources and establish a multi-level financial service system, while actively innovating digital financial service models to provide the necessary financial support for improving the performance of innovation and entrepreneurship; Third, there is an exemplary effect on the performance improvement of innovation among university, so we should strengthen the exchange and cooperation of innovation activities among university, build a university regional innovation linkage network, encourage human resource and information communication with each other. Fourth, the government should actively guide all social parties to participate and support the development of high-quality innovation projects, forming government policies and financial support, and social direct and indirect diversified financing methods to effectively relieve the constraints of innovation financing and better promote the performance of innovation and entrepreneurship. Accelerate the reform of the financial system, focus on improving financial efficiency and better play the intermediary role of financial efficiency. The lack of a perfect credit system is an important reason that hinders the development of digital finance. We should continuously improve and perfect the credit system of digital finance, promote the reform of the financial system, help the development of digital finance, promote the improvement of the performance of innovation and entrepreneurship, and boost the high-quality development of China's economy.

There are still some limitations in the research of this paper, and limited to the availability of data, this paper explores the research of digital finance on the performance of innovation at the city-university level. Along with the continuous establishment of micro-databases and the rise of big data mining technology, more micro- level data can be collected for empirical

research in the subsequent study. In addition, when exploring the paths of digital finance affecting the performance of innovation in universities, this paper conducts theoretical and empirical analysis from two paths: credit supply and financial efficiency and strives to achieve self-consistency between theory and empirical evidence. However, the impact of digital finance on the performance of innovation may be multi-channel, and limited by the space, data and focus of the research question, this paper does not explore the issue in more depth, which can be studied in the future.

## Supporting information

**S1 File.**
(PDF)

## Acknowledgments

The authors would like to thank the editors and reviewers for their helpful comments and suggestions.

## Author Contributions

**Methodology:** Lan Tao.

**Resources:** Song Tan, Lan Tao.

**Software:** Song Tan, Lan Tao.

**Writing – original draft:** Song Tan.

**Writing – review & editing:** Song Tan.

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
