## [Decision Letter · Decision Letter 0]

14 Feb 2023

PONE-D-22-31410Digital Finance and Performance of Innovation and EntrepreneurshipPLOS ONE

Dear Dr. Li,

Thank you for submitting your manuscript to PLOS ONE. After careful consideration, we feel that it has merit but does not fully meet PLOS ONE’s publication criteria as it currently stands. Therefore, we invite you to submit a revised version of the manuscript that addresses the points raised during the review process.

ACADEMIC EDITOR: Please read with attention the observations made by the reviewers and take all of them into account when revising your paper. I agree with the reviewers and consider that you should solve the weakness mentioned fact that would increase the quality of the paper.

We look forward to receiving your revised manuscript.

Kind regards,

Valentina Diana Rusu, PhD

Academic Editor

PLOS ONE

Journal Requirements:

We will update your Data Availability statement on your behalf to reflect the information you provide..

Reviewers' comments:

Reviewer's Responses to Questions

**Comments to the Author**

1. Is the manuscript technically sound, and do the data support the conclusions?

Reviewer #1: Partly

Reviewer #2: Yes

2. Has the statistical analysis been performed appropriately and rigorously? 

Reviewer #1: I Don't Know

Reviewer #2: Yes

3. Have the authors made all data underlying the findings in their manuscript fully available?

Reviewer #1: No

Reviewer #2: Yes

4. Is the manuscript presented in an intelligible fashion and written in standard English?

Reviewer #1: No

Reviewer #2: Yes

5. Review Comments to the Author

Reviewer #1: Dear authors, I enjoyed reading your paper However I have the following concerns:

1) the manuscript requires profound language editing. there are to many mistakes in the manuscript. in some parts it reads like being prepared in another language first and machine translated after that

2) the research question is hidden in the text. that should be made clear and placed visible in the text. the reason for this is that the reader needs some guidance to understand why the paper is written and what might be the value

3) the manuscript lacks a conceptual framework. chapter 2 describes a bit of digital finance, a bit of innovation and a bit of entrepreneurship. however it doesn't provide an overarching framework why and how these bits and pieces go together. instead you present 2 hypothesis which basically repeat the paper title. also I don't see much difference between H1 and H2

4) I doubt that the indicators chosen for DEA are suitable to describe innovation and entrepreneurship performance. indicators presented in table 1.

gien these weaknesses I also doubt the vailidity of conclusions. calcluations seem ok however the lacking conceptual framework and indicator selection respectively doesn't justify the analysis. the paper should first develop a profound concpetual framework and run econometric analysis second only. the variables used for the analysis require profound explanation.

Reviewer #2: First of all, let me congratulate the author on a very interesting study. Overall, this paper is well written and structured. The model seems to me nice and the results are quite good even if there some limitations that the author is aware about.

The survey data and statistical analysis are well presented and the results could be interesting but the article should have a review of literature and a discussion more related to the activity, more structured, in order to justify the methodological choices and to show the contribution of the study.

Some past researchers have been studied the issues. Comparing the previous studies, what is major implication for academicians and enterprisers? Author should be discussed in discussion and implications section. I recommend acceptance with minor revisions.

6. PLOS authors have the option to publish the peer review history of their article (what does this mean?). If published, this will include your full peer review and any attached files.

Reviewer #1: No

Reviewer #2: No

---

## [Author Response · Author response to Decision Letter 0]

29 May 2023

Dear Editors and Reviewers:

Thank you for your letter and the reviewers' comments concerning our manuscript entitled " Digital Finance and Performance in Innovation and Entrepreneurship: Evidence from Higher Education" (Manuscript ID:2231410). Those comments are valuable and helpful for improving our paper and the essential guiding significance to our research. We have substantially revised manuscripts comments by the reviews. Modified portions are marked in red in the article. The primary corrections in the form and the responses to the reviewer's comments as follow:

Answer to the reviewer's 1 comments:

Review comments 

Reviewer #1: Dear authors, I enjoyed reading your paper However I have the following concerns:

Question:

1) the manuscript requires profound language editing. there are to many mistakes in the manuscript. in some parts it reads like being prepared in another language first and machine translated after that.

Answer: I am so sorry about the language, to solve the problem we polished full paper into professional native speaker polishing intuition, and ask for help for native speaker, the certification of polishing as follow:

2) the research question is hidden in the text. that should be made clear and placed visible in the text. the reason for this is that the reader needs some guidance to understand why the paper is written and what might be the value.

Answer: thanks for the reviewer suggestion, the revise as follow:

In China, entrepreneurial talent is key to the development of an entrepreneurial economy. Hence, achieving internal development and cultivating innovative and entrepreneurial talent to meet current needs have become key to the reform and development of higher education (Weihui Mei 2020). However, innovation and entrepreneurship in colleges and universities is a high-risk activity that is associated with high costs and uncertain output, especially in an economic downturn. The financial constraints universities face is a key factor restricting the improvement of innovation technology and patents. Strategies to improve universities’ performance in innovation and entrepreneurship constitute a main research problem related to achieving internal development. 

3) the manuscript lacks a conceptual framework. chapter 2 describes a bit of digital finance, a bit of innovation and a bit of entrepreneurship. however it doesn't provide an overarching framework why and how these bits and pieces go together. instead you present 2 hypothesis which basically repeat the paper title. also I don't see much difference between H1 and H2.

Answer: thanks for the reviewer suggestion, the revise as follow:

Based on the existing literature and drawing on signalling theory, this study hypothesized that digital finance influences university-level innovation and entrepreneurship performance through the following two channel mechanisms. 

First, the development of digital finance can improve the quality of university-level innovation and entrepreneurship by expanding credit supply and relieving the financing constraints universities face (Pyka, A. 2017). Entrepreneurial returns are uncertain, and information asymmetry tends to favour capital owners. Potential entrepreneurs are bound to face financing constraint dilemmas, and university innovation activities are also affected by uncertain financing constraints (D. A. Harper 2020). Relying mainly on borrowers' collateral materials and financial information to decide whether to lend renders the decision-making basis too narrow, which contributes to the financing difficulty dilemma. The development of digital inclusive finance can facilitate the use of big data, machine learning, and other methods to collect and analyse a large number of borrowing individuals' bills, shopping records, and other information in order to expand the credit supply, relieve financing difficulties and expensive financing for universities (Lili Ding 2022), and help improve the quality of innovation and entrepreneurship. 

Second, the development of digital finance can also effectively improve financial efficiency and inject new momentum towards improving the quality of innovation and entrepreneurship (Carungu, J. 2021). Financial efficiency is an important dimension to measure financial institutions’ effectiveness at promoting the allocation of funds in economic activities. Digital finance’s greatest functions are to overcome temporal and spatial limitations and establish a direct point-to-point contact between financial product providers and demanders, thus accelerating the speed of capital circulation across time and space, which will, in turn, enhance financial efficiency and reduce financial enterprises’ operating costs (Ramlee, S. 2013). Tang, S. (2020) suggested that digital finance can also, to a certain extent, drive the reshaping of the traditional financial system, force the traditional financial sector to transform and upgrade, and improve the allocation efficiency and risk control ability of financial resources. Moreover, the development of digital finance lowers the financial services threshold, improves financial efficiency, and contributes significantly to regional total factor productivity. In summary, this study proposed the following research hypothesis.

H2: Digital finance can, by expanding the credit supply and improving financial efficiency, promote universities’ innovation and entrepreneurship performance.

According to the abovementioned literature, signalling theory fits well with the current study's objective regarding university-level digital finance and innovation performance. It is thought that digital finance can, by decreasing market moral hazards and enhancing the efficiency of financial services, improve universities’ innovation and entrepreneurship performance (Lili Ding 2022). Furthermore, digital finance lowers the financial services threshold, improves financial efficiency, and contributes significantly to regional total factor productivity (Ramlee, S. 2013). Based on the above discussion, the present study’s conceptual model is visualized in Fig. 1. 

 Fig. 1 Digital finance conceptual model

The above digital finance and innovation and entrepreneurship performance model encompasses digital finance, credit supply, and innovation behaviours in higher education. The figure shows two mechanisms. First, the development of digital inclusive finance can facilitate the use of big data and machine learning to collect and analyse a large amount of organizational borrowing information to expand the credit supply, relieve the financing difficulties and expensive financing universities face (Lili Ding, 2022), and help improve the quality of innovation and entrepreneurship. Secondly, the development of digital finance will lower the financial services threshold, improve financial efficiency, and significantly contribute to higher education innovation and entrepreneurship.

Although the present study has contributed to the accumulation of innovation and entrepreneurship education research, it has several limitations. First, the research lacks theoretical depth; specifically, it lacks scientific performance evaluation theoretical models, a reasonable index design, and comprehensive evaluation content. Second, the evaluation subjects were relatively homogenous as the study mainly focused on teachers and students. Third, regarding the research methods, the study was mainly qualitative, and the quantitative research lacks large data samples; consequently, the research conclusions are neither convincing nor universally applicable. Nevertheless, compared with the extant literature, we drew upon information theory and the two perspectives of credit supply and financial service efficiency to discuss the relationship of digital finance and university-level innovation and entrepreneurship performance with the aim of improving universities’ innovation development and addressing the problem of insufficient investment in university-level innovation and entrepreneurship. 

4) I doubt that the indicators chosen for DEA are suitable to describe innovation and entrepreneurship performance. indicators presented in table 1.

Answer:

I am so sorry about that, so I add some literature to support our indicators chosen, the revise as follow:

The independent variable was tertiary institutions’ innovation and entrepreneurship performance. The extant literature suggests the convenience of using human capital, financial investment, and research investment indicators to closely examine higher education innovation and entrepreneurship capabilities (María Leticia 2020). In this study, based on the existing literature (Yang Chen 2015), we divide the university inputs into three major inputs of human input, financial input and material input, and the university outputs into three major outputs of talent cultivation, scientific research, and social service. In terms of human input, considering that colleges and universities are the main places for knowledge production and talent cultivation, the main work of their workers can be divided into teaching and research, administration and auxiliary support. So according to the different types of work, this paper selects three indicators for the human input of colleges and universities: the number of full-time teachers, the number of administrative staff, and the number of teaching and auxiliary staff. In terms of financial input, this paper refers to the general division method of income sources of colleges and universities and divides the financial input of colleges and universities into three items: state financial allocation, education business income of colleges and universities and other income. Among them, the state financial appropriation includes education business appropriation, scientific research fund appropriation, other appropriation, etc.; the education business income of colleges and universities mainly refers to the tuition fee income of colleges and universities; other income refers to other non-financial and non-education business income of colleges and universities, such as business income, contribution from affiliated units and social donation. Material input. The material input of colleges and universities mainly includes three aspects: one is the input of fixed assets such as teaching buildings, libraries, student dormitories and canteens; the second is the input of library literature; the third is the input of instruments and equipment necessary for teaching and research. Accordingly, this paper selects 3 indicators of fixed assets formation, books and teaching and research instruments and equipment value to portray the physical investment of universities. The output indicators include talent cultivation, scientific research and social service. This paper will use DEA-Malmquist index to calculate the innovation and entrepreneurship performance indicators of colleges and universities, and the system of innovation and entrepreneurship performance indicators of colleges and universities is shown in Table 1.

Reviewer #2: First, let me congratulate the author on a very interesting study. Overall, this paper is well written and structured. The model seems to me nice and the results are quite good even if there some limitations that the author is aware about. The survey data and statistical analysis are well presented and the results could be interesting.

Question (1): but the article should have a review of literature and a discussion more related to the activity, more structured, to justify the methodological choices and to show the contribution of the study.

Answer: thanks for the reviewer suggestion, the revise as follow:

Although the present study has contributed to the accumulation of innovation and entrepreneurship education research, it has several limitations. First, the research lacks theoretical depth; specifically, it lacks scientific performance evaluation theoretical models, a reasonable index design, and comprehensive evaluation content. Second, the evaluation subjects were relatively homogenous as the study mainly focused on teachers and students. Third, regarding the research methods, the study was mainly qualitative, and the quantitative research lacks large data samples; consequently, the research conclusions are neither convincing nor universally applicable. Nevertheless, compared with the extant literature, we drew upon information theory and the two perspectives of credit supply and financial service efficiency to discuss the relationship of digital finance and university-level innovation and entrepreneurship performance with the aim of improving universities’ innovation development and addressing the problem of insufficient investment in university-level innovation and entrepreneurship. 

Question(2):Some past researchers have been studied the issues. Comparing the previous studies, what is major implication for academicians and enterprisers? Author should be discussed in discussion and implications section. I recommend acceptance with minor revisions.

Answer: thanks for the reviewer suggestion, the revise as follow:

This study examined the impact of digital finance on universities’ performance in innovation and entrepreneurship to draw useful insights and references to strengthen the theoretical research on and the practical application of innovation and entrepreneurship in universities through empirical methods. Compared with the existing literature, this paper’s contributions are as follows. We are the first to focus on the digital finance perspective. Specifically, we pioneered exploration of the relationship between digital finance and universities’ performance in innovation and entrepreneurship. Another contribution lies in the fact that this study empirically investigated the digital finance mechanism’s relationship with university-level performance in innovation and entrepreneurship. In this study, we reconstructed the university-level innovation and entrepreneurship performance index using the data envelopment analysis–Malmquist method. Finally, this paper provides policy recommendations to better assess the level of development of innovation and entrepreneurship in universities and improve national levels of innovation and technology. 

We tried our best to improve the manuscript and made full changes in the manuscript. These changes will not influence the content and framework of the paper. We appreciate Editors/Reviewers’ warm work earnestly and hope that the correction will meet with approval. Once again, thank you very much for your comments and suggestions.

Yours sincerely

---

## [Decision Letter · Decision Letter 1]

7 Jun 2023

PONE-D-22-31410R1Digital Finance and Performance of Innovation and Entrepreneurship：PLOS ONE

Dear Dr. tao,

Thank you for submitting your manuscript to PLOS ONE. After careful consideration, we feel that it has merit but does not fully meet PLOS ONE’s publication criteria as it currently stands. Therefore, we invite you to submit a revised version of the manuscript that addresses the points raised during the review process.

ACADEMIC EDITOR: one of the reviewers still has some concerns regarding your paper, and does not recommend the publication. Please take into account its recommendations, and change your paper accordingly. 

We look forward to receiving your revised manuscript.

Kind regards,

Valentina Diana Rusu, PhD

Academic Editor

PLOS ONE

Reviewers' comments:

Reviewer's Responses to Questions

**Comments to the Author**

1. If the authors have adequately addressed your comments raised in a previous round of review and you feel that this manuscript is now acceptable for publication, you may indicate that here to bypass the “Comments to the Author” section, enter your conflict of interest statement in the “Confidential to Editor” section, and submit your "Accept" recommendation.

Reviewer #1: (No Response)

Reviewer #2: All comments have been addressed

2. Is the manuscript technically sound, and do the data support the conclusions?

Reviewer #1: Partly

Reviewer #2: Yes

3. Has the statistical analysis been performed appropriately and rigorously? 

Reviewer #1: I Don't Know

Reviewer #2: Yes

4. Have the authors made all data underlying the findings in their manuscript fully available?

Reviewer #1: No

Reviewer #2: Yes

5. Is the manuscript presented in an intelligible fashion and written in standard English?

Reviewer #1: No

Reviewer #2: Yes

6. Review Comments to the Author

Reviewer #1: Thank you for revising the paper. However I still have concerns:

1) I don't find H1, the paper now begins with H2 page 13.

2) the innovation and entrepreneurship performance index used is questionable. The authors use variables which aren't reflecting the actual innovation and entrepreneurship related output. output indicators refer to talent development, scientific research and social services which is common practice. but which of these indicators is considered to explain the 1) innovation performance and 2) entrepreneurship performance? both indicators (or if you like the combined indicator) don't reflect innovation and entrepreneurship outcomes at all.

3) based especially on 2 I don't consider the econometric analysis being useful and vaild. the variables used in my view don't respond to the intended innovation and entrepreneurship performance. in other words the research framework isn't solid.

Reviewer #2: First of all, let me congratulate the author on a very interesting study. Overall, this paper is well written and structured. The model seems to me nice and the results are quite good even if there some limitations that the author is aware about.

Well dome with developed paper

7. PLOS authors have the option to publish the peer review history of their article (what does this mean?). If published, this will include your full peer review and any attached files.

Reviewer #1: No

Reviewer #2: No

---

## [Author Response · Author response to Decision Letter 1]

22 Jun 2023

Respond lists

Dear Editors and Reviewers: 

Thank you for your letter and the reviewers' comments concerning our manuscript entitled " Digital Finance and Performance in Innovation: Evidence from Universities" (Manuscript ID:2231410). Those comments are valuable and helpful for improving our paper and the essential guiding significance to our research. We have substantially revised manuscripts comments by the reviews. Modified portions are marked in red in the article. The primary corrections in the form and the responses to the reviewer's comments as follow: Answer to the reviewer's 1 comments: Review comments

Reviewer #1: Thank you for revising the paper. However, I still have concerns:1) I don't find H1, the paper now begins with H2 page 13.

Answer: thanks reviewer suggestion! I am so sorry about this mistake caused by polish article. We have already added the mission hypothesis 1 as following and rewritten sector 2:

2.1 Digital finance and innovation performance 

Innovation is one of the driving factors of economic and social development, and the Austrian economist Schumpeter's "innovation theory" defines innovation as a new combination of production factors. It is an effective way to cultivate innovative talents (Śledzik, Karol,2013). The training of innovative talents is inseparable from financial support, and there is an important economic relationship between digital finance and the performance of innovation and entrepreneurship in colleges and universities. 

In view of the uneven and insufficient development of traditional financial system, the long fund flow cycle of universities and the insufficient supply of funds to universities in less developed areas restrict the improvement of innovation and entrepreneurship performance (Hamburg, I,2017). The vigorous development of information technology has given birth to digital finance as an emerging financial service model. Digital inclusive finance has a positive impact on financial inclusion and stability, provides a basis for innovation, and can effectively increase the investment in innovation and entrepreneurship in universities. Combing the existing studies (Dykes, E,2013; Stefaniak, J,2019). 

The impact of digital inclusive finance on the performance of university innovation and entrepreneurship can be summarized as follows: digital finance can accurately assess the default risk of customers through big data modeling, and carry out real-time and dynamic supervision of credit customers, effectively control and identify risks, avoid adverse selection and moral risks in the financial market, improve risk control ability, explore more business space, and bring more new opportunities for university innovation and entrepreneurship .It brings more new opportunities for university innovation and entrepreneurship (Becker,2018). At the same time, digital finance also effectively reduces the cost of information search and matching between borrowers and lenders. Goldfarb & Tucker (2019) pointed out that big data can effectively reduce economic costs and help improve the performance of university innovation and entrepreneurship. Finally, the development of digital finance provides a good financial environment for university innovation and entrepreneurship. Trinugroho, I (2020) suggest that Digital inclusive finance has bred a distributed business pattern with strong inclusiveness and permeability, which is conducive to the communication and cooperation of innovation subjects in the distributed innovation network, constructing a collaborative innovation mechanism and helping to improve the quality of university innovation and entrepreneurship. 

In addition, the combination of traditional finance such as payment, investment and credit with digital technology effectively improves the convenience of customers using financial services and provides a good external environment for financial services to improve the performance of university innovation and entrepreneurship. In summary, this study proposed the following research hypothesis 1.

H1: Digital finance can effectively promote the improvement of innovation and entrepreneurship.

2) the innovation and entrepreneurship performance index used is questionable. The authors use variables which aren't reflecting the actual innovation and entrepreneurship related output. output indicators refer to talent development, scientific research and social services which is common practice. but which of these indicators is considered to explain the 1) innovation performance and 2) entrepreneurship performance? both indicators (or if you like the combined indicator) don't reflect innovation and entrepreneurship outcomes at all.

Answer: thanks reviewer suggestion! Regarding the selection of indicators is based on the literature for the selection of indicators. This paper adopts the reviewer's opinion that the selected indicators in this paper cannot respond to the entrepreneurial performance, first, change the innovation and entrepreneurship performance of universities into innovation performance; second, add the literature basis and support for the selection of innovation performance indicators, and rewrite the analysis and literature theory of innovation performance indicators selection still as following:

The index data related to colleges and universities used in this study were mainly obtained from the 2010–2019 Compilation of Basic Statistics on Colleges and Universities of the Ministry of Education, and the city-level data were obtained from the China City Statistical Yearbook for the year’s corresponding to the period of interest. The final sample comprised 72 colleges and universities’ 10-year panel data.

The independent variable was tertiary institutions’ innovation. The extant literature suggests the convenience of using human capital, financial investment, and research investment indicators to closely examine higher education innovation capabilities (María Leticia 2020). In this study, based on the existing literature (Yang Chen 2015), we divide the university inputs into three major inputs of human input, financial input, and material input.

Based on Xiong (2019), we measure university innovation performance outputs into three major outputs of talent development, scientific research, and Innovation education effects. In terms of human input, considering that colleges and universities are the main places for innovation knowledge production and innovation talent cultivation, the main work of their workers can be divided into innovation teaching and research, administration, and auxiliary support. 

So according to the different types of work, this paper selects three indicators for the innovation human input of colleges and universities: the number of full-time teachers, the number of administrative staff, and the number of teaching and auxiliary staff. In terms of financial input, innovation research outcomes in universities are closely linked to financial investment (Fu, Y. 2023). This paper refers to the general division method of income sources of colleges and universities and divides the financial input of colleges and universities into three items: state financial allocation, education business income of colleges and universities and other income. Among them, the state financial appropriation includes education business appropriation, scientific innovation research fund appropriation, other appropriation, etc.; the education business income of colleges and universities mainly refers to the tuition fee income of colleges and universities; other income refers to other non-financial and non-education business income of colleges and universities, such as innovation business income, contribution from affiliated units and social donation.

 Material input. Innovation education effects is an important indicator of innovation performance (Vincent-Lancrin, S. 2017). In his work propose a model of innovation performance where the probability of being innovative performance depends on the innovation education effects. They examine whether the technology transfer, patent and consultancy adopted can explain different types of innovation performance at universities. Dutta et al (2011) analyze the relationship between innovation material input and innovation education effects. Therefore, the innovation material input of colleges and universities mainly includes three aspects: one is the input of fixed assets such as innovation teaching buildings, libraries, student dormitories and canteens; the second is the input of library literature; the third is the input of instruments and equipment necessary for teaching and research. Accordingly, this paper selects 3 indicators of fixed assets formation, books and teaching and research instruments and equipment value to portray the physical investment of universities. The output indicators include talent cultivation, scientific research, and social service. 

3) based especially on 2 I don't consider the econometric analysis being useful and vaild. the variables used in my view don't respond to the intended innovation and entrepreneurship performance. in other words, the research framework isn't solid.

Answer: To address the problem that the empirical analysis does not reflect entrepreneurial performance, this paper accepts the opinion of the reviewers and changes the title of the paper to "Digital Finance and Performance in Innovation: Evidence from Higher Education ". The selection of innovation performance indicators is supported by the literature. Based on Xiong (2019), we measure university innovation performance outputs into three major outputs of talent development, scientific research, and Innovation education effects. In terms of human input, considering that colleges and universities are the main places for innovation knowledge production and innovation talent cultivation, the main work of their workers can be divided into innovation teaching and research, administration, and auxiliary support. 

So according to the different types of work, this paper selects three indicators for the innovation human input of colleges and universities: the number of full-time teachers, the number of administrative staff, and the number of teaching and auxiliary staff. In terms of financial input, innovation research outcomes in universities are closely linked to financial investment (Fu, Y. 2023). This paper refers to the general division method of income sources of colleges and universities and divides the financial input of colleges and universities into three items: state financial allocation, education business income of colleges and universities and other income. Among them, the state financial appropriation includes education business appropriation, scientific innovation research fund appropriation, other appropriation, etc.; the education business income of colleges and universities mainly refers to the tuition fee income of colleges and universities; other income refers to other non-financial and non-education business income of colleges and universities, such as innovation business income, contribution from affiliated units and social donation.

 Material input. Innovation education effects is an important indicator of innovation performance (Vincent-Lancrin, S. 2017). In his work propose a model of innovation performance where the probability of being innovative performance depends on the innovation education effects. They examine whether the technology transfer, patent and consultancy adopted can explain different types of innovation performance at universities. Dutta et al (2011) analyze the relationship between innovation material input and innovation education effects. Therefore, the innovation material input of colleges and universities mainly includes three aspects: one is the input of fixed assets such as innovation teaching buildings, libraries, student dormitories and canteens; the second is the input of library literature; the third is the input of instruments and equipment necessary for teaching and research. Accordingly, this paper selects 3 indicators of fixed assets formation, books and teaching and research instruments and equipment value to portray the physical investment of universities. The output indicators include talent cultivation, scientific research, and social service. 

Reference:

[1] Vincent-Lancrin, S., G. Jacotin, J. Urgel, S. Kar and C. González-Sancho (2017), Measuring Innovation in Education: A Journey to the Future, OECD Publishing, Paris.

[2] Dutta, D., Li, J., and Merenda, M. (2011). Fostering entrepreneurship: impact of specialization and diversity in education. International Entrepreneurship and Management Journal, 7(2):163–179.

[3] Fu, Y. (2023). The impact of government funding on research innovation: An empirical analysis of Chinese universities. Managerial and Decision Economics, 44(1). 285– 296.

[4] Xiong (2019). The construction of innovation and entrepreneurship teaching performance evaluation index system in colleges and universities and its application. Journal of Innovation and Enterprise Education.10(1): 674-893.

---

## [Decision Letter · Decision Letter 2]

5 Jul 2023

PONE-D-22-31410R2Digital Finance and Performance of Innovation: Evidence from UniversitiesPLOS ONE

Dear Dr. tao,

Thank you for submitting your manuscript to PLOS ONE. After careful consideration, we feel that it has merit but does not fully meet PLOS ONE’s publication criteria as it currently stands. Therefore, we invite you to submit a revised version of the manuscript that addresses the points raised during the review process. Reviewer 1 considers that you have not taken into account its recommendations and recommends the rejection of the paper from publication. 

We look forward to receiving your revised manuscript.

Kind regards,

Valentina Diana Rusu, PhD

Academic Editor

PLOS ONE

Reviewers' comments:

Reviewer's Responses to Questions

**Comments to the Author**

1. If the authors have adequately addressed your comments raised in a previous round of review and you feel that this manuscript is now acceptable for publication, you may indicate that here to bypass the “Comments to the Author” section, enter your conflict of interest statement in the “Confidential to Editor” section, and submit your "Accept" recommendation.

Reviewer #1: (No Response)

Reviewer #2: All comments have been addressed

2. Is the manuscript technically sound, and do the data support the conclusions?

Reviewer #1: Partly

Reviewer #2: Yes

3. Has the statistical analysis been performed appropriately and rigorously? 

Reviewer #1: I Don't Know

Reviewer #2: Yes

4. Have the authors made all data underlying the findings in their manuscript fully available?

Reviewer #1: Yes

Reviewer #2: Yes

5. Is the manuscript presented in an intelligible fashion and written in standard English?

Reviewer #1: Yes

Reviewer #2: (No Response)

6. Review Comments to the Author

Reviewer #1: the response to previous review round remarks aren't satisfactory. I strongly recommend to take the remarks serious and avoid informing that it's reviewers perspective.

Reviewer #2: First of all, let me congratulate the author on a very interesting study. Overall, this paper is well written and structured. The model seems to me nice and the results are quite good even if there some limitations that the author is aware about.

The survey data and statistical analysis are well presented and the results could be interesting but the article should have a review of literature and a discussion more related to the activity, more structured, in order to justify the methodological choices and to show the contribution of the study.

Some past researchers have been studied the issues. Comparing the previous studies, what is major implication for academicians and enterprisers? Author should be discussed in discussion and implications section.

7. PLOS authors have the option to publish the peer review history of their article (what does this mean?). If published, this will include your full peer review and any attached files.

Reviewer #1: No

Reviewer #2: No

---

## [Author Response · Author response to Decision Letter 2]

7 Jul 2023

Dear Editors and Reviewers:

Thank you for your letter and the reviewers' comments concerning our manuscript entitled " Digital Finance and Performance in Innovation: Evidence from universities " (Manuscript ID:2231410). After three major revisions, the article underwent major changes in terms of framework, economic theory, language native and selection of indicator with innovation performance. Three revisions are highly lighted with red color in manuscripts.

We have done a lot of work on these three revisions, and the three responses are shown below. After three comments from external reviewers, this paper was rewritten and improved from various aspects such as relevant literature support, index selection, theoretical logical framework, and economic theory. Finally, we hope that the editors will accept our article after several major revisions, thank you!

Respond lists 1

Dear Editors and Reviewers:

Thank you for your letter and the reviewers' comments concerning our manuscript entitled " Digital Finance and Performance in Innovation and Entrepreneurship: Evidence from Higher Education" (Manuscript ID:2231410). Those comments are valuable and helpful for improving our paper and the essential guiding significance to our research. We have substantially revised manuscripts comments by the reviews. Modified portions are marked in red in the article. The primary corrections in the form and the responses to the reviewer's comments as follow:

Answer to the reviewer's 1 comments:

Review comments 

Reviewer #1: Dear authors, I enjoyed reading your paper However I have the following concerns:

Question:

1) the manuscript requires profound language editing. there are to many mistakes in the manuscript. in some parts it reads like being prepared in another language first and machine translated after that.

Answer: I am so sorry about the language, to solve the problem we polished full paper into professional native speaker polishing intuition, and ask for help for native speaker, the certification of polishing as follow:

2) the research question is hidden in the text. that should be made clear and placed visible in the text. the reason for this is that the reader needs some guidance to understand why the paper is written and what might be the value.

Answer: thanks for the reviewer suggestion, the revise as follow:

In China, entrepreneurial talent is key to the development of an entrepreneurial economy. Hence, achieving internal development and cultivating innovative and entrepreneurial talent to meet current needs have become key to the reform and development of higher education (Weihui Mei 2020). However, innovation and entrepreneurship in colleges and universities is a high-risk activity that is associated with high costs and uncertain output, especially in an economic downturn. The financial constraints universities face is a key factor restricting the improvement of innovation technology and patents. Strategies to improve universities’ performance in innovation and entrepreneurship constitute a main research problem related to achieving internal development. 

3) the manuscript lacks a conceptual framework. chapter 2 describes a bit of digital finance, a bit of innovation and a bit of entrepreneurship. however it doesn't provide an overarching framework why and how these bits and pieces go together. instead you present 2 hypothesis which basically repeat the paper title. also I don't see much difference between H1 and H2.

Answer: thanks for the reviewer suggestion, the revise as follow:

Based on the existing literature and drawing on signalling theory, this study hypothesized that digital finance influences university-level innovation and entrepreneurship performance through the following two channel mechanisms. 

First, the development of digital finance can improve the quality of university-level innovation and entrepreneurship by expanding credit supply and relieving the financing constraints universities face (Pyka, A. 2017). Entrepreneurial returns are uncertain, and information asymmetry tends to favour capital owners. Potential entrepreneurs are bound to face financing constraint dilemmas, and university innovation activities are also affected by uncertain financing constraints (D. A. Harper 2020). Relying mainly on borrowers' collateral materials and financial information to decide whether to lend renders the decision-making basis too narrow, which contributes to the financing difficulty dilemma. The development of digital inclusive finance can facilitate the use of big data, machine learning, and other methods to collect and analyse a large number of borrowing individuals' bills, shopping records, and other information in order to expand the credit supply, relieve financing difficulties and expensive financing for universities (Lili Ding 2022), and help improve the quality of innovation and entrepreneurship. 

Second, the development of digital finance can also effectively improve financial efficiency and inject new momentum towards improving the quality of innovation and entrepreneurship (Carungu, J. 2021). Financial efficiency is an important dimension to measure financial institutions’ effectiveness at promoting the allocation of funds in economic activities. Digital finance’s greatest functions are to overcome temporal and spatial limitations and establish a direct point-to-point contact between financial product providers and demanders, thus accelerating the speed of capital circulation across time and space, which will, in turn, enhance financial efficiency and reduce financial enterprises’ operating costs (Ramlee, S. 2013). Tang, S. (2020) suggested that digital finance can also, to a certain extent, drive the reshaping of the traditional financial system, force the traditional financial sector to transform and upgrade, and improve the allocation efficiency and risk control ability of financial resources. Moreover, the development of digital finance lowers the financial services threshold, improves financial efficiency, and contributes significantly to regional total factor productivity. In summary, this study proposed the following research hypothesis.

H2: Digital finance can, by expanding the credit supply and improving financial efficiency, promote universities’ innovation and entrepreneurship performance.

According to the abovementioned literature, signalling theory fits well with the current study's objective regarding university-level digital finance and innovation performance. It is thought that digital finance can, by decreasing market moral hazards and enhancing the efficiency of financial services, improve universities’ innovation and entrepreneurship performance (Lili Ding 2022). Furthermore, digital finance lowers the financial services threshold, improves financial efficiency, and contributes significantly to regional total factor productivity (Ramlee, S. 2013). Based on the above discussion, the present study’s conceptual model is visualized in Fig. 1. 

 Fig. 1 Digital finance conceptual model

The above digital finance and innovation and entrepreneurship performance model encompasses digital finance, credit supply, and innovation behaviours in higher education. The figure shows two mechanisms. First, the development of digital inclusive finance can facilitate the use of big data and machine learning to collect and analyse a large amount of organizational borrowing information to expand the credit supply, relieve the financing difficulties and expensive financing universities face (Lili Ding, 2022), and help improve the quality of innovation and entrepreneurship. Secondly, the development of digital finance will lower the financial services threshold, improve financial efficiency, and significantly contribute to higher education innovation and entrepreneurship.

Although the present study has contributed to the accumulation of innovation and entrepreneurship education research, it has several limitations. First, the research lacks theoretical depth; specifically, it lacks scientific performance evaluation theoretical models, a reasonable index design, and comprehensive evaluation content. Second, the evaluation subjects were relatively homogenous as the study mainly focused on teachers and students. Third, regarding the research methods, the study was mainly qualitative, and the quantitative research lacks large data samples; consequently, the research conclusions are neither convincing nor universally applicable. Nevertheless, compared with the extant literature, we drew upon information theory and the two perspectives of credit supply and financial service efficiency to discuss the relationship of digital finance and university-level innovation and entrepreneurship performance with the aim of improving universities’ innovation development and addressing the problem of insufficient investment in university-level innovation and entrepreneurship. 

4) I doubt that the indicators chosen for DEA are suitable to describe innovation and entrepreneurship performance. indicators presented in table 1.

Answer:

I am so sorry about that, so I add some literature to support our indicators chosen, the revise as follow:

The independent variable was tertiary institutions’ innovation and entrepreneurship performance. The extant literature suggests the convenience of using human capital, financial investment, and research investment indicators to closely examine higher education innovation and entrepreneurship capabilities (María Leticia 2020). In this study, based on the existing literature (Yang Chen 2015), we divide the university inputs into three major inputs of human input, financial input and material input, and the university outputs into three major outputs of talent cultivation, scientific research, and social service. In terms of human input, considering that colleges and universities are the main places for knowledge production and talent cultivation, the main work of their workers can be divided into teaching and research, administration and auxiliary support. So according to the different types of work, this paper selects three indicators for the human input of colleges and universities: the number of full-time teachers, the number of administrative staff, and the number of teaching and auxiliary staff. In terms of financial input, this paper refers to the general division method of income sources of colleges and universities and divides the financial input of colleges and universities into three items: state financial allocation, education business income of colleges and universities and other income. Among them, the state financial appropriation includes education business appropriation, scientific research fund appropriation, other appropriation, etc.; the education business income of colleges and universities mainly refers to the tuition fee income of colleges and universities; other income refers to other non-financial and non-education business income of colleges and universities, such as business income, contribution from affiliated units and social donation. Material input. The material input of colleges and universities mainly includes three aspects: one is the input of fixed assets such as teaching buildings, libraries, student dormitories and canteens; the second is the input of library literature; the third is the input of instruments and equipment necessary for teaching and research. Accordingly, this paper selects 3 indicators of fixed assets formation, books and teaching and research instruments and equipment value to portray the physical investment of universities. The output indicators include talent cultivation, scientific research and social service. This paper will use DEA-Malmquist index to calculate the innovation and entrepreneurship performance indicators of colleges and universities, and the system of innovation and entrepreneurship performance indicators of colleges and universities is shown in Table 1.

Reviewer #2: First, let me congratulate the author on a very interesting study. Overall, this paper is well written and structured. The model seems to me nice and the results are quite good even if there some limitations that the author is aware about. The survey data and statistical analysis are well presented and the results could be interesting.

Question (1): but the article should have a review of literature and a discussion more related to the activity, more structured, to justify the methodological choices and to show the contribution of the study.

Answer: thanks for the reviewer suggestion, the revise as follow:

Although the present study has contributed to the accumulation of innovation and entrepreneurship education research, it has several limitations. First, the research lacks theoretical depth; specifically, it lacks scientific performance evaluation theoretical models, a reasonable index design, and comprehensive evaluation content. Second, the evaluation subjects were relatively homogenous as the study mainly focused on teachers and students. Third, regarding the research methods, the study was mainly qualitative, and the quantitative research lacks large data samples; consequently, the research conclusions are neither convincing nor universally applicable. Nevertheless, compared with the extant literature, we drew upon information theory and the two perspectives of credit supply and financial service efficiency to discuss the relationship of digital finance and university-level innovation and entrepreneurship performance with the aim of improving universities’ innovation development and addressing the problem of insufficient investment in university-level innovation and entrepreneurship. 

Question(2):Some past researchers have been studied the issues. Comparing the previous studies, what is major implication for academicians and enterprisers? Author should be discussed in discussion and implications section. I recommend acceptance with minor revisions.

Answer: thanks for the reviewer suggestion, the revise as follow:

This study examined the impact of digital finance on universities’ performance in innovation and entrepreneurship to draw useful insights and references to strengthen the theoretical research on and the practical application of innovation and entrepreneurship in universities through empirical methods. Compared with the existing literature, this paper’s contributions are as follows. We are the first to focus on the digital finance perspective. Specifically, we pioneered exploration of the relationship between digital finance and universities’ performance in innovation and entrepreneurship. Another contribution lies in the fact that this study empirically investigated the digital finance mechanism’s relationship with university-level performance in innovation and entrepreneurship. In this study, we reconstructed the university-level innovation and entrepreneurship performance index using the data envelopment analysis–Malmquist method. Finally, this paper provides policy recommendations to better assess the level of development of innovation and entrepreneurship in universities and improve national levels of innovation and technology. 

We tried our best to improve the manuscript and made full changes in the manuscript. These changes will not influence the content and framework of the paper. We appreciate Editors/Reviewers’ warm work earnestly and hope that the correction will meet with approval. Once again, thank you very much for your comments and suggestions.

Yours sincerely 

a Response to Reviewers letter 2

Dear Editors and Reviewers: 

Thank you for your letter and the reviewers' comments concerning our manuscript entitled " Digital Finance and Performance in Innovation: Evidence from Universities" (Manuscript ID:2231410). Those comments are valuable and helpful for improving our paper and the essential guiding significance to our research. We have substantially revised manuscripts comments by the reviews. Modified portions are marked in red in the article. The primary corrections in the form and the responses to the reviewer's comments as follow: Answer to the reviewer's 1 comments: Review comments

1. Reviewer #1: Thank you for revising the paper. However, I still have concerns:1) I don't find H1, the paper now begins with H2 page 13.

Answer: thanks reviewer suggestion! I am so sorry about this mistake caused by polish article. We have already added the mission hypothesis 1 as following and rewritten sector 2:

2.1 Digital finance and innovation performance 

Innovation is one of the driving factors of economic and social development, and the Austrian economist Schumpeter's "innovation theory" defines innovation as a new combination of production factors. It is an effective way to cultivate innovative talents (Śledzik, Karol,2013). The training of innovative talents is inseparable from financial support, and there is an important economic relationship between digital finance and the performance of innovation and entrepreneurship in colleges and universities. 

In view of the uneven and insufficient development of traditional financial system, the long fund flow cycle of universities and the insufficient supply of funds to universities in less developed areas restrict the improvement of innovation and entrepreneurship performance (Hamburg, I,2017). The vigorous development of information technology has given birth to digital finance as an emerging financial service model. Digital inclusive finance has a positive impact on financial inclusion and stability, provides a basis for innovation, and can effectively increase the investment in innovation and entrepreneurship in universities. Combing the existing studies (Dykes, E,2013; Stefaniak, J,2019). 

The impact of digital inclusive finance on the performance of university innovation and entrepreneurship can be summarized as follows: digital finance can accurately assess the default risk of customers through big data modeling, and carry out real-time and dynamic supervision of credit customers, effectively control and identify risks, avoid adverse selection and moral risks in the financial market, improve risk control ability, explore more business space, and bring more new opportunities for university innovation and entrepreneurship .It brings more new opportunities for university innovation and entrepreneurship (Becker,2018). At the same time, digital finance also effectively reduces the cost of information search and matching between borrowers and lenders. Goldfarb & Tucker (2019) pointed out that big data can effectively reduce economic costs and help improve the performance of university innovation and entrepreneurship. Finally, the development of digital finance provides a good financial environment for university innovation and entrepreneurship. Trinugroho, I (2020) suggest that Digital inclusive finance has bred a distributed business pattern with strong inclusiveness and permeability, which is conducive to the communication and cooperation of innovation subjects in the distributed innovation network, constructing a collaborative innovation mechanism and helping to improve the quality of university innovation and entrepreneurship. 

In addition, the combination of traditional finance such as payment, investment and credit with digital technology effectively improves the convenience of customers using financial services and provides a good external environment for financial services to improve the performance of university innovation and entrepreneurship. In summary, this study proposed the following research hypothesis 1.

H1: Digital finance can effectively promote the improvement of innovation and entrepreneurship.

2. the innovation and entrepreneurship performance index used is questionable. The authors use variables which aren't reflecting the actual innovation and entrepreneurship related output. output indicators refer to talent development, scientific research and social services which is common practice. but which of these indicators is considered to explain the 1) innovation performance and 2) entrepreneurship performance? both indicators (or if you like the combined indicator) don't reflect innovation and entrepreneurship outcomes at all.

Answer: thanks reviewer suggestion! Regarding the selection of indicators is based on the literature for the selection of indicators. This paper adopts the reviewer's opinion that the selected indicators in this paper cannot respond to the entrepreneurial performance, first, change the innovation and entrepreneurship performance of universities into innovation performance; second, add the literature basis and support for the selection of innovation performance indicators, and rewrite the analysis and literature theory of innovation performance indicators selection still as following:

The index data related to colleges and universities used in this study were mainly obtained from the 2010–2019 Compilation of Basic Statistics on Colleges and Universities of the Ministry of Education, and the city-level data were obtained from the China City Statistical Yearbook for the year’s corresponding to the period of interest. The final sample comprised 72 colleges and universities’ 10-year panel data.

The independent variable was tertiary institutions’ innovation. The extant literature suggests the convenience of using human capital, financial investment, and research investment indicators to closely examine higher education innovation capabilities (María Leticia 2020). In this study, based on the existing literature (Yang Chen 2015), we divide the university inputs into three major inputs of human input, financial input, and material input.

Based on Xiong (2019), we measure university innovation performance outputs into three major outputs of talent development, scientific research, and Innovation education effects. In terms of human input, considering that colleges and universities are the main places for innovation knowledge production and innovation talent cultivation, the main work of their workers can be divided into innovation teaching and research, administration, and auxiliary support. 

So according to the different types of work, this paper selects three indicators for the innovation human input of colleges and universities: the number of full-time teachers, the number of administrative staff, and the number of teaching and auxiliary staff. In terms of financial input, innovation research outcomes in universities are closely linked to financial investment (Fu, Y. 2023). This paper refers to the general division method of income sources of colleges and universities and divides the financial input of colleges and universities into three items: state financial allocation, education business income of colleges and universities and other income. Among them, the state financial appropriation includes education business appropriation, scientific innovation research fund appropriation, other appropriation, etc.; the education business income of colleges and universities mainly refers to the tuition fee income of colleges and universities; other income refers to other non-financial and non-education business income of colleges and universities, such as innovation business income, contribution from affiliated units and social donation.

 Material input. Innovation education effects is an important indicator of innovation performance (Vincent-Lancrin, S. 2017). In his work propose a model of innovation performance where the probability of being innovative performance depends on the innovation education effects. They examine whether the technology transfer, patent and consultancy adopted can explain different types of innovation performance at universities. Dutta et al (2011) analyze the relationship between innovation material input and innovation education effects. Therefore, the innovation material input of colleges and universities mainly includes three aspects: one is the input of fixed assets such as innovation teaching buildings, libraries, student dormitories and canteens; the second is the input of library literature; the third is the input of instruments and equipment necessary for teaching and research. Accordingly, this paper selects 3 indicators of fixed assets formation, books and teaching and research instruments and equipment value to portray the physical investment of universities. The output indicators include talent cultivation, scientific research, and social service. 

3.based especially on 2 I don't consider the econometric analysis being useful and vaild. the variables used in my view don't respond to the intended innovation and entrepreneurship performance. in other words, the research framework isn't solid.

Answer: To address the problem that the empirical analysis does not reflect entrepreneurial performance, this paper accepts the opinion of the reviewers and changes the title of the paper to "Digital Finance and Performance in Innovation: Evidence from Higher Education ". The selection of innovation performance indicators is supported by the literature. Based on Xiong (2019), we measure university innovation performance outputs into three major outputs of talent development, scientific research, and Innovation education effects. In terms of human input, considering that colleges and universities are the main places for innovation knowledge production and innovation talent cultivation, the main work of their workers can be divided into innovation teaching and research, administration, and auxiliary support. 

So according to the different types of work, this paper selects three indicators for the innovation human input of colleges and universities: the number of full-time teachers, the number of administrative staff, and the number of teaching and auxiliary staff. In terms of financial input, innovation research outcomes in universities are closely linked to financial investment (Fu, Y. 2023). This paper refers to the general division method of income sources of colleges and universities and divides the financial input of colleges and universities into three items: state financial allocation, education business income of colleges and universities and other income. Among them, the state financial appropriation includes education business appropriation, scientific innovation research fund appropriation, other appropriation, etc.; the education business income of colleges and universities mainly refers to the tuition fee income of colleges and universities; other income refers to other non-financial and non-education business income of colleges and universities, such as innovation business income, contribution from affiliated units and social donation.

 Material input. Innovation education effects is an important indicator of innovation performance (Vincent-Lancrin, S. 2017). In his work propose a model of innovation performance where the probability of being innovative performance depends on the innovation education effects. They examine whether the technology transfer, patent and consultancy adopted can explain different types of innovation performance at universities. Dutta et al (2011) analyze the relationship between innovation material input and innovation education effects. Therefore, the innovation material input of colleges and universities mainly includes three aspects: one is the input of fixed assets such as innovation teaching buildings, libraries, student dormitories and canteens; the second is the input of library literature; the third is the input of instruments and equipment necessary for teaching and research. Accordingly, this paper selects 3 indicators of fixed assets formation, books and teaching and research instruments and equipment value to portray the physical investment of universities. The output indicators include talent cultivation, scientific research, and social service. 

Reference:

[1] Vincent-Lancrin, S., G. Jacotin, J. Urgel, S. Kar and C. González-Sancho (2017), Measuring Innovation in Education: A Journey to the Future, OECD Publishing, Paris.

[2] Dutta, D., Li, J., and Merenda, M. (2011). Fostering entrepreneurship: impact of specialization and diversity in education. International Entrepreneurship and Management Journal, 7(2):163–179.

[3] Fu, Y. (2023). The impact of government funding on research innovation: An empirical analysis of Chinese universities. Managerial and Decision Economics, 44(1). 285– 296.

[4] Xiong (2019). The construction of innovation and entrepreneurship teaching performance evaluation index system in colleges and universities and its application. Journal of Innovation and Enterprise Education.10(1): 674-893.

a Response to Reviewers’1 letter 3 revise

Dear Editors and Reviewers: 

Thank you for your letter and the reviewers' 1 comments concerning our manuscript entitled " Digital Finance and Performance in Innovation: Evidence from Universities" (Manuscript ID:2231410). 

The reviewers’1 made a total of three comments in the second revision, mainly that the paper lacks research hypothesis1, the unreasonable selection of innovation performance indicators in universities and the research framework problems caused by the unreasonable selection of indicators. In response to these three main problems. Considering reviewer 1 not satisfied with our second revise, based on the second revision we make extensive revisions, including replacing indicators of university innovation performance based on extensive literature, re-measuring innovation performance indices in third revise manuscripts, in the hope of meeting reviewer requirements. Modified portions are marked in red in the article. The third corrections in the form and the responses to the reviewer 1 comments as follow: 

Reviewer #1: Thank you for revising the paper. However, I still have concerns:1) I don't find H1, the paper now begins with H2 page 13.

Answer: Research hypothesis 1 that digital finance drives innovation performance in universities has been added to research hypothesis 2, the economics of information theory, in the second revision, and the authors believe that this revision satisfies the first missing research hypothesis 1 raised by the reviewer1, revise parts seen as paper part 2 with red highlights , The revision is as follows 

2.1 Digital finance and innovation performance 

Innovation is one of the driving factors of economic and social development, and the Austrian economist Schumpeter's "innovation theory" defines innovation as a new combination of production factors. It is an effective way to cultivate innovative talents (Śledzik, Karol,2013). The training of innovative talents is inseparable from financial support, and there is an important economic relationship between digital finance and the performance of innovation and entrepreneurship in colleges and universities. 

In view of the uneven and insufficient development of traditional financial system, the long fund flow cycle of universities and the insufficient supply of funds to universities in less developed areas restrict the improvement of innovation and entrepreneurship performance (Hamburg, I,2017). The vigorous development of information technology has given birth to digital finance as an emerging financial service model. Digital inclusive finance has a positive impact on financial inclusion and stability, provides a basis for innovation, and can effectively increase the investment in innovation and entrepreneurship in universities. Combing the existing studies (Dykes, E,2013; Stefaniak, J,2019). 

The impact of digital inclusive finance on the performance of university innovation and entrepreneurship can be summarized as follows: digital finance can accurately assess the default risk of customers through big data modeling, and carry out real-time and dynamic supervision of credit customers, effectively control and identify risks, avoid adverse selection and moral risks in the financial market, improve risk control ability, explore more business space, and bring more new opportunities for university innovation and entrepreneurship .It brings more new opportunities for university innovation and entrepreneurship (Becker,2018). At the same time, digital finance also effectively reduces the cost of information search and matching between borrowers and lenders. Goldfarb & Tucker (2019) pointed out that big data can effectively reduce economic costs and help improve the performance of university innovation and entrepreneurship. Finally, the development of digital finance provides a good financial environment for university innovation and entrepreneurship. Trinugroho, I (2020) suggest that Digital inclusive finance has bred a distributed business pattern with strong inclusiveness and permeability, which is conducive to the communication and cooperation of innovation subjects in the distributed innovation network, constructing a collaborative innovation mechanism and helping to improve the quality of university innovation and entrepreneurship. 

In addition, the combination of traditional finance such as payment, investment and credit with digital technology effectively improves the convenience of customers using financial services and provides a good external environment for financial services to improve the performance of university innovation and entrepreneurship. In summary, this study proposed the following research hypothesis 1.

H1: Digital finance can effectively promote the improvement of innovation and entrepreneurship.

2. the innovation and entrepreneurship performance index used is questionable. The authors use variables which aren't reflecting the actual innovation and entrepreneurship related output. output indicators refer to talent development, scientific research and social services which is common practice. but which of these indicators is considered to explain the 1) innovation performance and 2) entrepreneurship performance? both indicators (or if you like the combined indicator) don't reflect innovation and entrepreneurship outcomes at all.

Answer: In response to the second indicator selection issue raised by the reviewer, the following changes are made in this paper: First, the whole paper only studies innovation performance, but not entrepreneurial performance. Second, in terms of the selection of indicators for innovation performance, we read a large amount of relevant literature and combined it with the current literature to finally select innovation talent, innovation scientific research, and innovation education effects to reflect We finally selected three indicators to reflect innovation performance of universities and redid the DEA calculation and benchmark regression calculation. The authors concluded that the modified indicators can reflect the innovation performance of universities after deleting the entrepreneurial performance of universities, reading a large amount of relevant literature, and giving the literature basis for the selection of innovation performance indicators of universities. Meanwhile, this paper redoes the calculation of the new college innovation performance index and benchmark regression. Finally, the authors believe that the new college innovation index and college innovation performance are closely related, both can reflect the degree of college innovation performance and can answer the doubts of reviewer 1.

The revise innovation performance indicators at section 3.1 and table 1 with red highlights as following:

To empirically test the effect of digital finance on universities’ innovation performance, this study constructed an econometric model of the influence of digital finance on the quality of innovation in universities. This section will explain the reason that innovation performance input and output data sources, measuring of innovation performance and baseline model.

3.1 Data sources

The independent variable was innovation performance. The extant literature suggests the convenience of using human capital, financial investment, and education Material indicators to closely examine higher education innovation capabilities (María Leticia 2020) through DEA analysis. Hence, in this study, based on the existing literature (Yang Chen 2015), we divide the university innovation inputs into three major inputs of Innovation Human Capital, Financial investment of Innovation Education, and Innovation Education Material in Table 1. 

In terms of outputs indicators for innovation performance, this paper based on Xiong (2019), to measure university innovation performance outputs into three major outputs of innovation talent, innovation scientific research and innovation education effects. Considering that colleges and universities are the main places for innovation knowledge production and innovation talent cultivation, the main work of their workers can be divided into number of innovation activities administrative staff and innovation teaching and support staff two items, we must keep in mind that the capacity a university must innovate depends on talent assets and knowledge it possesses (Subramaniam & Youndt, 2005),which means this perspective of innovation performance depends on innovation human capital, such as innovation activities administrative staff and innovation teaching and support staff (Fu, Y. 2023).

In terms of financial investment of innovation education. This paper refers to the general division method of income sources of colleges and universities and divides the financial input of colleges and universities into three items: state innovation education financial allocation, innovation education business income and other innovation activities income. Among them, the state innovation education financial allocation includes education scientific innovation research fund appropriation; the innovation education business income of colleges and universities mainly refers to the tuition fee income of innovation colleges; other innovation activities income refers to other non-financial and non-education business innovation income of colleges and universities, such as innovation business income, contribution from innovation affiliated units(Okoye.2022). Considering that financial investment of innovation education play a critical role in innovation development, especially for the development of innovation research and patent promotion in universities (UN ,2021).

Innovation Education Material. Innovation education effects is an important indicator of innovation performance (Vincent-Lancrin, S. 2017). In his work propose a model of innovation performance where the probability of being innovative performance depends on the innovation education effects. They examine whether the technology transfer, patent and consultancy adopted can explain different types of innovation performance at universities. Dutta et al (2011) analyze the relationship between innovation material input and innovation education effects. 

Therefore, the innovation material input of colleges and universities mainly includes three aspects: The value of innovation research instruments and equipment, number of books about innovation research, and fixed asset of innovation research. Based on numerous literatures (Dosi ,2005; Marta ,2019) , this paper choose three output indicators include innovation talent, innovation scientific research, and innovation education effects to reflect innovation performance of universities.

 According to Marta (2019), we measure innovation performance from universities through DEA-Malmquist methods. Research has addressed the application of the DEA-Malmquist in innovation performance at universities, considering non-profit organizations focused on innovation production rather than profitability (Al-Hosaini & Sofian, 2015). These measures respond better to universities innovation performance, and the system of innovation performance indicators of colleges and universities is shown in Table 1.

Table 1: Construction of innovation performance index system in universities

Input Indicators General

(8 items)

Innovation Human

Capital 1 Number of innovation activities

administrative staff Y

 2 Number of innovation teaching and support staff Y

Financial investment of Innovation Education 3 State innovation education financial allocation Y

 4 Innovation Education Business Income Y

 5 Other innovation activities income Y

Innovation Education Material 6 The value

Of Innovation research instruments and Equipment Y

 7 Number of books about innovation research Y

 8 Fixed asset of innovation research Y

Output Indicators General

(11 items)

Innovation Talent 

 1 Innovative entrepreneurial Projects Y

 2 Changjiang Scholar Award Program Y

 3 Number of International Students Y

Innovation Scientific Research 4 Number of Patent conversions Y

 5 Number of published papers (web of science) Y

 6 Number of Patents Y

 7 Number of innovating awards Y

Innovation Education Effects 8 Number of consultancy studies adopted Y

 9 Technology transfer contract amount Y

 10 Total amount of patent sales Y

 11 Number of further education and training Y

Fig. 1 Digital finance conceptual model

Based on Kubak, M (2020), considering universities is non-profit organizations, innovation performance must reflect the dynamic innovation change status of relative innovation efficiency, therefore we adopt Malmquist index to better analysis innovation performance chance from panel data.

3.based especially on 2 I don't consider the econometric analysis being useful and valid. the variables used in my view don't respond to the intended innovation and entrepreneurship performance. in other words, the research framework isn't solid.

Answer: The reviewers believe that the selected indicators do not reflect the innovation performance of universities, and therefore consider that the measurement of this paper does not have economic implications.

 The authors hereby declare that the framework of this paper is based on the theory of economics of information, with research hypothesis 1 and research hypothesis 2. Research hypothesis 2 is consistent with the mechanisms that follow, all within the framework of the theory of economics of information. Second, the selection of indicators is not reasonable. The authors in the second revision are reading a large amount of relevant literature and combining the opinions of the review experts1 to delete the entrepreneurial performance of colleges and universities and only study the innovation performance of colleges and universities. 

Thirdly, combining with the current published scholars' articles, the indicators are further selected and explained, and why the selected indicators can reflect the innovation performance of colleges and universities.

Fourth, since the second revision reviewer 1 was not satisfied, we read the relevant literature again, combined with the current published literature, re-selected the indicators of university innovation performance (see the second answer), and recalculated the university innovation performance index and the baseline regression model. Combining the above three answers and revisions, this paper concludes that the article measurement is economically meaningful, and the index selection can reflect the level of innovation performance of universities.

Reference:

[1] Vincent-Lancrin, S., G. Jacotin, J. Urgel, S. Kar and C. González-Sancho (2017), Measuring Innovation in Education: A Journey to the Future, OECD Publishing, Paris.

[2] Dutta, D., Li, J., and Merenda, M. (2011). Fostering entrepreneurship: impact of specialization and diversity in education. International Entrepreneurship and Management Journal, 7(2):163–179.

[3] Fu, Y. (2023). The impact of government funding on research innovation: An empirical analysis of Chinese universities. Managerial and Decision Economics, 44(1). 285– 296.

[4] Xiong (2019). The construction of innovation and entrepreneurship teaching performance evaluation index system in colleges and universities and its application. Journal of Innovation and Enterprise Education.10(1): 674-893.

[5] Guerrero, M., Urbano, D., Fayolle, A. et al (2016). Entrepreneurial universities: emerging models in the new social and economic landscape[J]. Small Bus Econ 47:551–563. 

https://doi.org/10.1007/s11187-016-9755-4.

[6] Kubak, M., et al. (2019). "On the Technical Efficiency and Productivity Changes Development in Slovakian Institutions of Higher Education 2005–2015: Malmquist Index Approach." Transformations in Business and Economics 18.

[7] Okoye, K., et al (2022). "Impact of global government investment on education and research development: A comparative analysis and demystifying the science, technology, innovation, and education conundrum." Global Transitions 4: 11-27.

[8] Lili Ding, Zhongchao Zhao, Lei Wang. (2022) Does online teaching strategy matter: exploring the effect of online teaching strategies on students’ ambidextrous innovation capacities based on the online teaching situation in China. Journal of Research on Technology in Education 0:0, pages 1-24.

[9] D. A. Harper, F.-F. Muñoz and F. J. Vázquez (2021). Innovation in online higher-education services: building complex systems[J]. Economics of Innovation and New Technology 30 (4): 412-431.

[10] Achtenhagen L, Naldi L, Melin L (2010). “Business growth” - do practitioners and scholars really talk about the same thing? [J]. Entrepreneurship Theory and Practice.34: 289–316. 10.1111/j.1540-6520.2010. 00376.x.

Finally, in response to the three questions raised by reviewer 1, the authors have made numerous revisions to the article in terms of theoretical framework and selection of indicators from the perspective of literature support and economic theory after reading a large amount of relevant literature. The authors believe that the currently selected output indicators can reflect the innovation performance of universities and deeply hope that the third revision will satisfy the reviewer 1. Thank you!

---

## [Decision Letter · Decision Letter 3]

31 Jul 2023

Digital Finance and Performance in Innovation: Evidence from Universities

PONE-D-22-31410R3

Dear Dr. tao,

We’re pleased to inform you that your manuscript has been judged scientifically suitable for publication and will be formally accepted for publication once it meets all outstanding technical requirements.

Kind regards,

Valentina Diana Rusu, PhD

Academic Editor

PLOS ONE

Additional Editor Comments (optional):

Reviewers' comments:

Reviewer's Responses to Questions

**Comments to the Author**

1. If the authors have adequately addressed your comments raised in a previous round of review and you feel that this manuscript is now acceptable for publication, you may indicate that here to bypass the “Comments to the Author” section, enter your conflict of interest statement in the “Confidential to Editor” section, and submit your "Accept" recommendation.

Reviewer #1: All comments have been addressed

2. Is the manuscript technically sound, and do the data support the conclusions?

Reviewer #1: Yes

3. Has the statistical analysis been performed appropriately and rigorously? 

Reviewer #1: Yes

4. Have the authors made all data underlying the findings in their manuscript fully available?

Reviewer #1: Yes

5. Is the manuscript presented in an intelligible fashion and written in standard English?

Reviewer #1: Yes

6. Review Comments to the Author

Reviewer #1: the manuscript improved signficantly during revisions. it's now in a shape which in my view justifies publication

7. PLOS authors have the option to publish the peer review history of their article (what does this mean?). If published, this will include your full peer review and any attached files.

Reviewer #1: No

---

## [Editor Report · Acceptance letter]

11 Sep 2023

PONE-D-22-31410R3 

Digital Finance and Performance in Innovation: Evidence from Universities 

Dear Dr. tao:

I'm pleased to inform you that your manuscript has been deemed suitable for publication in PLOS ONE. Congratulations! Your manuscript is now with our production department. 

Kind regards, 

on behalf of

Dr. Valentina Diana Rusu 

Academic Editor

PLOS ONE